# FreeGaussian: Guidance-free Controllable 3D Gaussian Splats with Flow Derivatives

## Abstract

Reconstructing controllable Gaussian splats from monocular video is a challenging task due to its inherently insufficient constraints. Widely adopted approaches supervise complex interactions with additional masks and control signal annotations, limiting their real-world applications. In this paper, we propose an annotation guidance-free method, dubbed **FreeGaussian**, that mathematically derives dynamic Gaussian motion from optical flow and camera motion using novel dynamic Gaussian constraints. By establishing a connection between 2D flows and 3D Gaussian dynamic control, our method enables annotation-free optimization and continuity of dynamic Gaussian motions from flow priors. Furthermore, we introduce a 3D spherical vector controlling scheme, which represents the state with a 3D Gaussian trajectory, thereby eliminating the need for complex 1D control signal calculations and simplifying controllable Gaussian modeling. Quantitative and qualitative evaluations on extensive experiments demonstrate the state-of-the-art visual performance and control capability of our method. Project page: https://freegaussian.github.io.

## 1 Introduction

Controllable view synthesis (CVS) aims to recover the 3D structure and interactable motions of a scene given a set of input views, which has garnered significant attention in various research fields, including content creation (Liao et al., 2024; Tang et al., 2023; Gao et al., 2024b), virtual reality (Steuer, 1992; Kerbl et al., 2023a; Waisberg et al., 2023) and robotic simulator (Huang et al., 2023; Qu et al., 2024; Lou et al., 2024). Mainstream methods Yu et al. (2023a); Fridovich-Keil et al. (2023) have recently achieved high-quality real-time rendering via 3D Gaussian representation Kerbl et al. (2023b) and extended to scene-level using large-scale annotated datasets (Qu et al., 2024).

Despite the impressive advances, a significant obstacle remains: the severe dependence on manual annotations hinders the practical application of mainstream methods. Existing methods either segment Gaussian ellipsoids in interactive regions via mask-based reprojection Yu et al. (2023a) or input control signals to jointly model neural radiance fields Kania et al. (2022); Fridovich-Keil et al. (2023); Qu et al. (2024). Without mask or control signal supervision in the training data, the model collapses, failing to decode the feature to color and losing scene control capabilities. Manual annotation guidance such as mask and control signal has become an indispensable and stringent condition for existing methods and datasets.

To address this challenge, we propose **FreeGaussian**, a guidance-free but effective Gaussian splatting method for controllable scene reconstruction, which automatically explores interactable structures and restores controllable scenes from successive frames, without any manual annotations. Our novel insight is that *dynamic Gaussian flow under instantaneous motion can be analytically derived from optical flow and camera motion via differential analysis*. It enables us to track dynamic Gaussian motion solely relying on camera views in the training process, which allows for localizing controllable structures and providing continuous optimization constraints. This innovation streamlines existing controllable view synthesis methods by introducing flow-based priors, eliminating the need for annotations and improving their real-world applicability.

More specifically, in the training stage, FreeGaussian directly derive dynamic Gaussians flow from 2D image optical flow and camera-induced camera flow, accumulated with Gaussian projection

displacements. By tracking the dynamic Gaussian flow, we highlight interactive dynamic 3DGS and obtain their trajectories via DBSCAN clustering, eliminating the dependence on manual mask annotations. To overcome the reliance on 1D control signal inputs, we introduce a **3D spherical vector controlling scheme** that exploits 3D Gaussian scene representations bypassing dynamic Gaussian trajectories as state representations, aligning with the splatting rasterization pipeline and greatly simplifying the control process. In constant, given the 3D control vector as input, the Gaussian dynamics are retrieved from the network during the control stage. Beyond localizing interactive Gaussians, the **dynamic Gaussian flow constraints** 3DGS motion between frames, guaranteeing smooth motion and eliminating ghosting artifacts to improve rendering quality. To the end, we implement the differentiable dynamic Gaussian flow analysis and constraints in CUDA, and evaluate the effectiveness of the 3D spherical vector controlling scheme on both synthetic and real-world datasets.

Extensive evaluations show that our method outperforms existing methods significantly in both novel view synthesis and scene controlling, enabling more accurate and efficient modeling of interactable content with no annotations. Contributions can be summarized as follows:

- We propose **FreeGaussian**, a novel annotations guidance-free Gaussian Splatting method for controllable scene reconstruction, which automatically explores interactable scene structures with flow priors, and restores scene interactivity without any manual annotations.
- FreeGaussian analytically derive the **dynamic Gaussian flow constraints** via differential analysis with alpha composition, which draws the mathematical link among optical flow, camera motion, and dynamic Gaussian flow. With the CUDA implementation, we leverage the flow constraints to refine Gaussian optimization, enabling unsupervised interactable scene structure localization and continuous Gaussian motion variation training.
- Exploiting 3D Gaussian explicitness, we introduce a **3D spherical vector controlling scheme**, avoiding traditional complex 1D control variable calculations bypassing 3DGS trajectory as state representation, further simplifying and accelerating interactive Gaussian modeling.

## 2 RELATED WORK

**4D Novel View Synthesis.** Neural Radiance Fields (NeRF) (Mildenhall et al., 2020) has innovated great progress in dynamic scene reconstruction. The existing methods can be categorized into three primary categories: time-varying, deformable-canonical, and hybrid representation methods. The time-varying methods (Du et al., 2021; Fang et al., 2022; Li et al., 2021; Park et al., 2021a; Pumarola et al., 2021; Tretschk et al., 2021; Yuan et al., 2021) directly model the radiance field over time and enhance the temporal information with time embedding, scene flow and *etc*. While, the deformable-canonical methods(Gao et al., 2021; Li et al., 2022; Park et al., 2021b; Xian et al., 2021) decouple the 4D field into dynamic deformable fields and static canonical spaces, querying canonical features by warped coordinates. In contrast, hybrid representation methods (Shao et al., 2023; Fridovich-Keil et al., 2023; Cao & Johnson, 2023; Song et al., 2023) have achieved high-quality reconstruction and fast rendering by exploiting time-space feature planes, dynamic voxels, and 4D hash encoding.

In contrast to fitting complex dynamic scenes with MLPs, 3D Gaussians Splatting (Kerbl et al., 2023b) has emerged as a popular choice recently, owing to the superior training efficiency and ultra-high-quality rendering speeds. Related progress typically learn dense Gaussian movements (Yang et al., 2023; Luiten et al., 2024) directly, leverage feature planes (Wu et al., 2023) or learnable motion basis (Kratimenos et al., 2023) for better rendering quality, or introduce flow loss (Guo et al., 2024) to enhancing different paradigms of dynamic 3DGS. More recently, S4D (He et al., 2024) introduced a generalized streaming pipeline that leverages Gaussians and 3D control points to reconstruct 4D real-world scenes.

**Controllable Scene Representation.** Decoupling color, occupancy, geometry from time provides increased flexibility over 4D reconstruction, with significant implications for digital humans (Rivero et al., 2024; Liu et al., 2023) and simulators (Qu et al., 2024; Wang et al., 2024). CoNeRF (Kania et al., 2022) pioneered this effort by extending HyperNeRF (Park et al., 2021b) and regressing the attribute and the mask to enable few-shot attribute control. CoNFies (Yu et al., 2023b) propose a controllable representation for face self-portraits by utilizing AU intensities and facial landmarks. EditableNeRF (Zheng et al., 2023) introduces detection key points and joint weights optimization. In

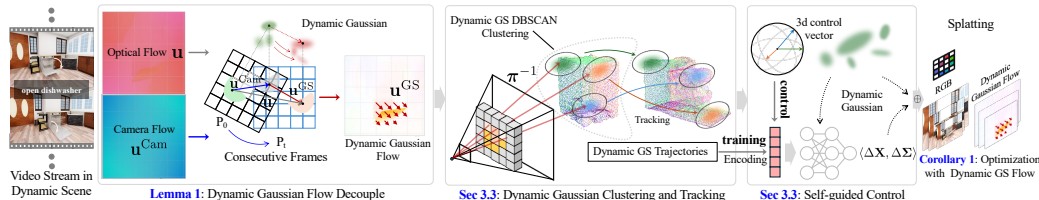

Figure 1: The overview of FreeGaussian. Given a set of video stream $\{\mathbf{P}(t), \mathbf{I}(t)\}$, our method recover controllable 3D Gaussians $\mathbf{G}^*$ with two stages. First, we pre-train a deformable 3DGS and calculate dynamic Gaussian flow $\mathbf{u}^{\text{GS}}$ from optical and camera flow with eq. (3). Then, we reproject dynamic Gaussian flow maps and cluster the highlight 3DGS with the DBSCAN algorithm, followed with trajectory calculation. In the controllable Gaussian training stage, we optimize Gaussians $\mathbf{G}$ and network $\boldsymbol{\Theta}$ using rasterization-based loss function in eq. (8), which measures the discrepancy between rendered images and input images, as well as dynamic Gaussian flows.

contrast, CoGS (Yu et al., 2023a) leveraged 3D Gaussians (Kerbl et al., 2023a) to achieve real-time control of dynamic scenes without requiring explicit control signals. More recently, LiveScene (Qu et al., 2024) advance the progress to scene-level and introduces an efficient factorization to decompose the interactive space. Despite their breakthroughs, these methods either require dense manual interaction variable annotations or mask supervision, limiting their real-world applicability.

## 3 METHODOLOGY

Figure. 1 shows the complete pipeline of FreeGaussian, which exploits the connections between dynamic Gaussian flow, optical flow, and camera motion, restoring scene interactivity without any manual annotations. The dynamic Gaussian flow derivative facilitates 3DGS trajectory clustering and enables a flexible 3D spherical vector control pipeline, which streamlines and accelerates the interactive Gaussian modeling scheme.

Hence, after recalling basic 3DGS preliminary in Section. 3.1, we draw the mathematical link among optical flow, camera motion, and dynamic Gaussian flow in Section. 3.2. With the dynamic Gaussian flow, we introduce the 3D spherical vector controlling scheme in Section. 3.3, which explores dynamic Gaussians and extracts their trajectories for joint training. The overall pipeline in Figure. 1 is optimized with loss function formulations in Section. 3.4.

### 3.1 PRELIMINARY OF 3DGS RASTERIZATION

3D Gaussian Splatting Kerbl et al. (2023b) (3DGS) explicitly represents scenes with millions of Gaussians and emerges ultra high-quality rendering performance recently. Given a set of images capture with corresponding camera poses, 3DGS models scenes by learning a set of 3D Gaussians $\mathbf{G} = \{G_i : (\mathbf{X}_i, \boldsymbol{\Sigma}_i, \mathbf{o}_i, \mathbf{H}_i)|i = 1, ..., N\}$, where $\mathbf{X}_i \in \mathbb{R}^3$, $\boldsymbol{\Sigma}_i \in \mathbb{R}^{3 \times 3}$, $\mathbf{o}_i \in \mathbb{R}$, and $\mathbf{H}_i \in \mathbb{R}^{48}$ are the center position, 3D covariance, opacity, and spherical harmonics of the $i$-th Gaussian, respectively. With the rasterization pipeline, 3DGS projects $\mathbf{G}$ to image planes as 2D Gaussians $\mathbf{g} = \{g_i : (\boldsymbol{\mu}_i, \boldsymbol{\Sigma}'_i, \mathbf{o}_i, \mathbf{c}_i)|i = 1, ..., N\}$ and blender pixel colors $\hat{\mathbf{C}}$ via alpha composition:

$$\hat{\mathbf{C}} = \sum_{i=1}^{N} \mathbf{c}_i \alpha_i T_i, \quad T_i = \prod_{j=1}^{i-1}(1 - \alpha_j), \tag{1}$$

where $\boldsymbol{\mu}_i \in \mathbb{R}^2$, $\boldsymbol{\Sigma}'_i \in \mathbb{R}^{2 \times 2}$, $\mathbf{c}_i \in \mathbb{R}^3$, $\alpha_i \in [0, 1]$ and $T_i \in [0, 1]$ are the 2d center, 2d covariance, color, alpha value and transmittance of 2D Gaussian $g_i$. The alpha value $\alpha_i$ at pixel coordinate $\mathbf{m}$ can be obtained by:

$$\alpha_i = \mathbf{o}_i \exp(-\frac{1}{2}(\mathbf{m} - \boldsymbol{\mu}_i)^T \boldsymbol{\Sigma}'^{-1}_i (\mathbf{m} - \boldsymbol{\mu}_i)). \tag{2}$$

With the supervision of observations, 3DGS optimizes parameters to minimize the photometric loss between rendered and ground-truth images.

### 3.2 DYNAMIC GAUSSIAN FLOW ANALYSIS

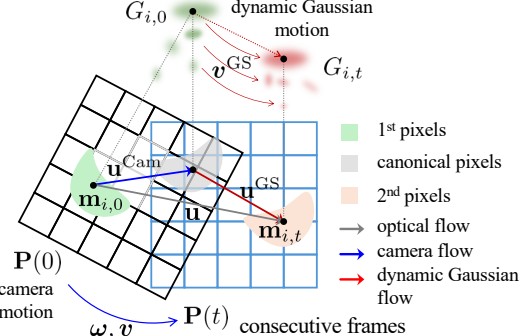

Figure 2: **Dynamic Gaussian flow illustration.** In interactive scenes, consider an instantaneous motion model, where the camera and 3D Gaussian hold separate velocities in consecutive frames. The projected optical flow $\mathbf{u}$ can be decomposed into camera flow $\mathbf{u}^{\text{Cam}}$ and dynamic Gaussian flow $\mathbf{u}^{\text{GS}}$, as described in eqs. (3) and (4).

Our insight is that dynamic Gaussian flow under instantaneous motion can be analytically decoupled from optical flow and camera motion via differential analysis with alpha composition. Considering a dynamic scene with interactive objects as shown in Figure. 2, the camera and 3D Gaussians hold separate velocities in consecutive frames $0$ and $t$. Assuming a dynamic 3D Gaussian $G_i$ with velocity $\boldsymbol{v}^{\text{GS}}$, it is projected as image measurement $g_i$ under the constant camera instantaneous motion by translation velocity $\boldsymbol{v}$ and rotational velocity $\boldsymbol{\omega}$. The optical flow $\mathbf{u}$ induced by $(\boldsymbol{v}, \boldsymbol{\omega})$ of a pixel $\mathbf{m} = (x, y)^{\top}$ can be obtained by *Lemma 1*:

**Lemma 1:** *Dynamic Gaussian flow $\mathbf{u}^{GS}$ under instantaneous motion can be derived from optical flow $\mathbf{u}$ and camera flow $\mathbf{u}^{Cam}$ with the following transform eq. (3).*

$$\mathbf{u} = \mathbf{u}^{\text{Cam}} + \mathbf{u}^{\text{GS}} + \boldsymbol{\Delta},$$

$$\mathbf{u}^{\text{Cam}} = \frac{\mathbf{A}\boldsymbol{v}}{Z} + \mathbf{B}\boldsymbol{\omega}, \quad \mathbf{u}^{\text{GS}} = \mathbf{A}\sum_{i=1}^{M} T_i \alpha_i \frac{\boldsymbol{v}^{\text{GS}}}{Z_i}, \quad \boldsymbol{\Delta} = \mathbf{A}\sum_{i=1}^{M} T_i \alpha_i \boldsymbol{v}(\frac{1}{Z_i} - \frac{1}{Z}),$$

$$\mathbf{A} = \begin{bmatrix} -f_x & 0 & x - c_x \\ 0 & -f_y & y - c_y \end{bmatrix}, \quad \mathbf{B} = \begin{bmatrix} \frac{(x-c_x)(y-c_y)}{f_y} & -f_x - \frac{(x-c_x)^2}{f_x} & \frac{(y-c_y)f_x}{f_y} \\ f_y + \frac{(y-c_y)^2}{f_y} & -\frac{(x-c_x)(y-c_y)}{f_x} & -\frac{(x-c_x)f_y}{f_x} \end{bmatrix},$$

(3)

where $f_x, f_y, c_x, c_y$ are camera intrinsics, $M$ denotes the number of Gaussian projections sorted with Gaussian depth $Z_i$ intersecting the pixel $\mathbf{m}$. Flow residual term $\boldsymbol{\Delta}$ are preserved to guarantee accuracy, even when they approach zero after refined optimization.

*Proof.* The proof involves analyzing camera motion and dynamic Gaussian motion under instantaneous motions. By differentiating the dynamic Gaussian center $\mathbf{X}_i$ and projection matrix in successive camera views $\mathbf{P}(0)$ and $\mathbf{P}(t)$, we derive the connection between dynamic Gaussian flow $\mathbf{u}_i^{\text{GS}}$, camera velocities $(\boldsymbol{v}, \boldsymbol{\omega})$, and optical flow $\mathbf{u}$. With alpha composition, we weight the flow with $w_i = \frac{T_i \alpha_i}{\Sigma_i T_i \alpha_i}$, and proof the mathematical relation described in eq. (3). Detailed derivation can be found in the supplementary material Section. 6. $\square$

The expression eq. (3) elucidates the triadic relationship, yet Gaussian flow is not amenable to joint 3DGS training. For flexibility, we consider a pixel $\mathbf{m}_{i,t}$ following 2D Gaussian distribution $g_i$ at time $t$, and obtain $\mathbf{m}_{i,t} \sim \mathcal{N}(\boldsymbol{\mu}_{i,t}, \boldsymbol{\Sigma}'_{i,t})$, with 2D mean $\boldsymbol{\mu}_{i,t}$ and covariance $\boldsymbol{\Sigma}'_{i,t} = \mathbf{B}_{i,t} \mathbf{B}_{i,t}^{\top}$. The following *Corollary* describes the dynamic Gaussian flow with 2D Gaussian means.

**Corollary 1:** *The dynamic Gaussian flow $\tilde{\mathbf{u}}^{GS}$ on image plane can be accumulated with 2D Gaussian means displacement $\boldsymbol{\mu}_{i,t} - \boldsymbol{\mu}_{i,0}$.*

$$\mathbf{u} = \mathbf{u}^{\text{Cam}} + \tilde{\mathbf{u}}^{\text{GS}} + \boldsymbol{\Delta}, \quad \tilde{\mathbf{u}}^{\text{GS}} = \sum_{i=1}^{M} T_i \alpha_i (\boldsymbol{\mu}_{i,t} - \boldsymbol{\mu}_{i,0}). \tag{4}$$

*Proof.* Assuming the Gaussian to be isotropic Gao et al. (2024b), with covariance matrix $\mathbf{B}_{i,t} \mathbf{B}_{i,t}^{\top} = \mathbf{R} \mathbf{S} \mathbf{S}^{\top} \mathbf{R}^{\top} = \sigma^2 \mathbf{I}$. With a constant instantaneous-motion model, the tiny varation of scaling factor $\sigma$ of each Gaussian can be simply ignored, and $\mathbf{B}_{i,t} \mathbf{B}_{i,0}^{-1} \approx \mathbf{I}$. Therefore, the projection flow of a dynamic Gaussian $G_i$ varying from $0$ to $t$ can be formulated as $\tilde{\mathbf{u}}_i^{\text{GS}} = \boldsymbol{\mu}_{i,t} - \boldsymbol{\mu}_{i,0}$. The difference between two Gaussian-distributed variables $\mathbf{m}_{i,0}$ and $\mathbf{m}_{i,t}$ can be expressed as:

$$\tilde{\mathbf{u}}_i^{\text{GS}} = \mathbf{x}_{i,t} - \mathbf{x}_{i,0} = \mathbf{B}_{i,t} \mathbf{B}_{i,0}^{-1} (\mathbf{x}_0 - \boldsymbol{\mu}_{i,t}) + \boldsymbol{\mu}_{i,t} - \mathbf{x}_0 = \boldsymbol{\mu}_{i,t} - \boldsymbol{\mu}_{i,0}. \tag{5}$$

By weighting the flow on both side, and substituting the flow into eq. (3), we obtain the relation among the optical flow, camera flow, and dynamic Gaussian flow. $\square$

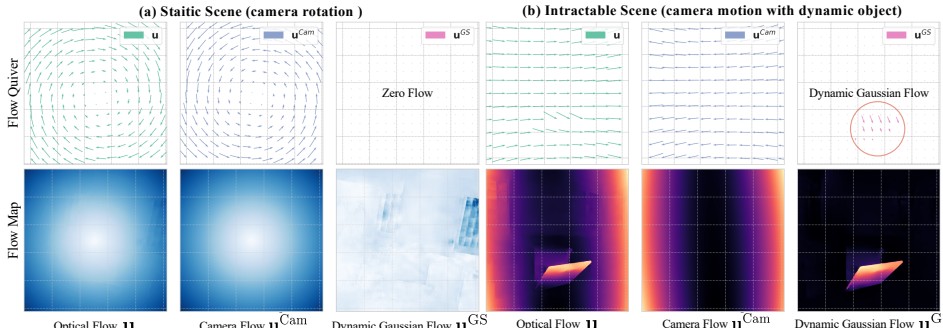

Figure 3: Illustration of dynamic Gaussian flow map under static and dynamic scenes. a) In static scenes with camera motion only, eq. (4) degenerate to pure camera flow and resulting zero dynamic Gaussian flow. b) In contrast, for dynamic scenes with interactive objects, the dynamic Gaussian flow map will highlight interactive 3D Gaussians.

Note that the isotropic Gaussian assumption helps to reduce computational complexity and enhance optimization stability. It is a common practice in many works (Gao et al., 2024a; Ling et al., 2024; Keetha et al., 2024). Nevertheless, it is still flexible to extend to anisotropic in practice with eq. (5).

**Discussion.** The expression in eqs. (3) and (4) reveals dynamic gaussian flow can be directly derived from 2D image flow $\mathbf{u}$ and camera-induced camera flow $\mathbf{u}^{\text{Cam}}$, accumeulated with 2DGS projection displacement $\boldsymbol{\mu}_{i,t} - \boldsymbol{\mu}_{i,0}$. This naturally aligns with the 3D Gaussian rasterization pipeline, providing continuous motion constraints for dynamic Gaussian optimization. Besides, in static Gaussian scenes, the equation degenerates to camera flow with $\mathbf{u} = \mathbf{u}^{\text{Cam}}$. Hence, the resulting dynamic Gaussian flow map will highlight interactive 3D Gaussians, as illustrated in Figure. 3.

## 3.3 SELF-GUIDED CONTROL WITH DYNAMIC 3DGS

Based on the discussion in Section. 3.2, dynamic Gaussian flow constraint eq. (4) provides continuous Gaussian constraints and, critically, exposes the position of interactive areas, whose changing topological structures in dynamic scenes are reflected in varying Gaussian. To overcome the severe dependence on mask annotations in existing methods, we propose leveraging dynamic Gaussian flow to explore dynamic Gaussians of interactive objects and extract their trajectories for joint training:

**Dynamic Gaussian clustering and tracking.** With the formulations in eq. (4), we pretrain a deformable 3DGS $\mathbf{G}'$ with a set of camera streams first. Then dynamic Gaussian flow $\mathbf{u}^{\text{GS}}$ from eq. (4) can be extracted frame-by-frame and binaried to obtain flow maps. By back-projecting the flow maps to identify dynamic 3D Gaussians, we highlight Gaussians $\mathcal{D} = \{g_i \mid i = 1, 2, \ldots, Q\}$ with sharp dynamics, as illustrated in Figure. 1. Next, we use unsupervised clustering algorithm **DBSCAN** to group dynamic Gaussians into clusters $\mathcal{C} = \{c_i \mid i = 1, 2, \ldots, K\}$, where $K$ is the number of interactive objects. The cluster centers evolve over time, generating continuous trajectories $\boldsymbol{\varsigma}(t, k)$, where $k$ indexing which objects the trajectory belongs to.

**3D Spherical Vector Control.** Conventional methods using a 1D state variable to describe object state changes are limited by the reliance on prior knowledge or Gaussian trajectory fitting, and their inability to accurately capture dynamic changes. We overcome these limitations by representing the Gaussian states with 3D spherical vectors, which can be directly obtained from dynamic Gaussian tracking trajectory. This technique eliminates the requirement of control signals and curve fitting while increasing control flexibility.

Specifically, in the training stage, we represent the Gaussian dynamics state using cluster trajectory coordinates $\mathbf{v}_c^i = \boldsymbol{\varsigma}(t, k) - \boldsymbol{\varsigma}(0, k)$, concatenated with Gaussian centers $\mathbf{X}_i$. Then, we encode the coordinates with $\mathbf{E}(\mathbf{v}_c^i, \mathbf{X}_i)$ and jointly train the model $\Theta$ to recover Gaussian dynamics $\langle \Delta \mathbf{X}_i, \Delta \boldsymbol{\Sigma}_i \rangle$:

$$\boldsymbol{f}_{\Theta}\left( \mathbf{E}(\mathbf{v}_c^i, \mathbf{X}_i) \right) \mapsto \langle \Delta \mathbf{X}_i, \Delta \boldsymbol{\Sigma}_i \rangle . \tag{6}$$

Then, we perform splatting rasterization in eq. (1) with the Gaussian combining with predicted dynamics. In contrast, during the control stage, we manually input interactive 3D vector $\mathbf{v}_c'$, retrieving the Gaussian dynamics from the network by $\boldsymbol{f}_{\Theta}(\mathbf{X}_i, \mathbf{v}_c')$.

### 3.4 LOSS FUNCTIONS

**Loss with dynamic Gaussian flow.** The expression in eq. (4) suggests that incorporating optical flow and camera flow prior to the loss function can improve 3DGS optimization and maintain dynamic Gaussian smooth transitions between frames. Hence, we propose a dynamic Gaussian flow loss $\mathcal{L}_{\text{uGS}}$ to optimize the dynamic Gaussian field $\mathbf{G}$ and network $\Theta$ with the following formulation:

$$\mathcal{L}_{\text{uGS}} = \left\| \mathbf{u} - \mathbf{u}^{\text{Cam}} - \sum_{i=1}^{M} T_i \alpha_i (\boldsymbol{\mu}_{i,t} - \boldsymbol{\mu}_{i,0}) \right\|^2, \tag{7}$$

where $\mathbf{u}$ and $\mathbf{u}^{\text{Cam}}$ can be calculated with optical flow estimator Contributors (2021) and eq. (4), respectively. Dynamic Gaussians $\mathbf{G}$ and $\Theta$ are optimized via the proposed dynamic gaussian flow supervision $\mathcal{L}_{\text{uGS}}$ in eq. (7) with the fundamental per-frame photometric supervision $\mathcal{L}_{\text{RGB}}$, and $\mathcal{L}_{\text{D-SSIM}}$. The loss function for FreeGaussian optimization can be formulated as:

$$\mathcal{L} = \lambda \mathcal{L}_{\text{RGB}} + (1 - \lambda) \mathcal{L}_{\text{D-SSIM}} + \beta \mathcal{L}_{\text{uGS}}. \tag{8}$$

## 4 EXPERIMENT

### 4.1 EXPERIMENTAL SETUP

**Datasets.** To evaluate the performance of FreeGaussian, we leverage the object level CoNeRF Synthetic and CoNeRF Controllable datasets in (Kania et al., 2022), and the scene level OmniSim and InterReal datasets in (Qu et al., 2024). Following (Qu et al., 2024), we divide the OmniSim and InterReal datasets into (#easy, #medium, #challenging) and (#medium, #challenging) respectively. No annotations are used in the training process.

**Baselines.** Three categories of sota baselines are compared, including 3D novel view synthesis methods (Mildenhall et al., 2020; Müller et al., 2022; Kerbl et al., 2023b;a), 4D deformable methods (Fridovich-Keil et al., 2023; Park et al., 2021b;a), and controllable scene reconstruction methods (Kania et al., 2022; Yu et al., 2023a; Fridovich-Keil et al., 2023; Qu et al., 2024). We conduct comprehensive evaluations of FreeGaussian from novel view synthesis in Section. 4.2, controllable rendering in Section. 4.2, and efficiency in Section. 4.3.

**Implementation details.** FreeGaussian is implemented based on nerfstudio (Tancik et al., 2023) and gsplat (Ye et al., 2024). We use RAFT Teed & Deng (2020); Contributors (2021) for optical flow prediction and perform DBSCAN clustering from dynamic Gaussian flow with Euclidean metric, $\epsilon = 0.05$ and minimal samples = 5. The cluster center corresponding to each Gaussian is encoded with hash grids and decoded with an 8-layer MLP with 256 neurons. The model is trained on an NVIDIA GeForce RTX 4090 GPU for 60k steps, using Adam optimizer with learning rate $1.6e^{-4}$ and batch size 1. The coarse-to-fine training process lasts 30 minutes and is divided into 3 stages, including 500 steps of canonical warmup, 30k steps 4D deformable training, and 30k steps of full training. For all experiments, we set loss weights of $\mathcal{L}_{\text{RGB}}$, $\mathcal{L}_{\text{D-SSIM}}$, and $\mathcal{L}_{\text{uGS}}$ as $\lambda = 0.8$, $(1 - \lambda) = 0.2$, and $\beta = 0.5$, respectively.

### 4.2 EVALUATION OF NOVEL VIEW SYNTHESIS

**Results on CoNeRF Synthetic and Controllable Datasets.** The quantitative results of our approach on the CoNeRF Synthetic and Controllable scenes are presented in Table. 1. Notably, our method surpasses all existing approaches in terms of PSNR, SSIM, and LPIPS metrics on CoNeRF Synthetic scenes, with a slight advantage over the second-best method, which benefits from dense labels. Furthermore, on CoNeRF Controllable scenes, our method attains the highest PSNR of 33.247, while demonstrating comparable SSIM and LPIPS scores to the SOTA methods. These results underscore the success of the guidance-free paradigm.

**Metric on OmniSim Dataset.** Table. 2 shows that FreeGaussian achieves the highest scores in PSNR, SSIM, and LPIPS on #medium subset of OmniSim, with optimal average scores of 33.249, 0.969, and 0.074, respectively. Specifically, our method surpasses sparse-label guidance methods Kania et al. (2022); Yu et al. (2023a) by nearly 1 dB in terms of PSNR. Although our approach

Table 1: **Quantitative results on CoNeRF synthetic and controllable datasets.** FreeGaussian tops the leaderboard on synthetic scenes and achieves the best PSNR on the controllable dataset.

| Method | CoNeRF Synthetic | | | CoNeRF Controllable | | |
|---|---|---|---|---|---|---|
| | PSNR↑ | SSIM↑ | LPIPS↓ | PSNR↑ | SSIM↑ | LPIPS↓ |
| NeRF (Mildenhall et al., 2020) | 25.299 | 0.843 | 0.197 | 28.795 | 0.951 | 0.210 |
| InstantNGP (Müller et al., 2022) | 27.057 | 0.903 | 0.230 | 26.391 | 0.884 | 0.278 |
| 3DGS (Kerbl et al., 2023a) | 32.576 | 0.977 | 0.077 | 25.945 | 0.834 | 0.414 |
| HyperNeRF(Park et al., 2021b) | 25.963 | 0.854 | 0.158 | 32.520 | 0.981 | 0.169 |
| K-Planes (Fridovich-Keil et al., 2023) | 33.301 | 0.933 | 0.150 | 31.811 | 0.912 | 0.262 |
| CoNeRF-$\mathcal{M}$(Kania et al., 2022) | 27.868 | 0.898 | 0.155 | 32.061 | 0.979 | 0.167 |
| CoNeRF(Kania et al., 2022) | 32.394 | 0.972 | 0.139 | 32.342 | 0.981 | 0.168 |
| CoGS (Yu et al., 2023a) | 33.455 | 0.960 | 0.064 | 32.601 | **0.983** | **0.164** |
| LiveScene Qu et al. (2024) | 43.349 | 0.986 | **0.011** | 32.782 | 0.932 | 0.186 |
| FreeGaussian (Ours) | **43.939** | **0.993** | **0.011** | **33.247** | 0.941 | 0.218 |

Table 2: **Quantitative results on OmniSim Dataset**. FreeGaussian surpasses prior works in most metrics, achieving the highest average scores for both the #medium subset and the entire dataset.

| Method | #Easy Sets | | | #Medium Sets | | | #Avg (all 20 Sets) | | |
|---|---|---|---|---|---|---|---|---|---|
| | PSNR↑ | SSIM↑ | LPIPS↓ | PSNR↑ | SSIM↑ | LPIPS↓ | PSNR↑ | SSIM↑ | LPIPS↓ |
| NeRF (Mildenhall et al., 2020) | 25.817 | 0.906 | 0.167 | 25.645 | 0.928 | 0.138 | 25.776 | 0.916 | 0.153 |
| InstantNGP (Müller et al., 2022) | 25.704 | 0.902 | 0.183 | 25.627 | 0.930 | 0.140 | 25.706 | 0.914 | 0.164 |
| HyperNeRF (Park et al., 2021b) | 30.708 | 0.908 | 0.316 | 31.621 | 0.936 | 0.265 | 30.748 | 0.917 | 0.299 |
| K-Planes (Fridovich-Keil et al., 2023) | 32.841 | 0.952 | 0.093 | 32.548 | 0.954 | 0.100 | 32.573 | 0.952 | 0.097 |
| CoNeRF (Kania et al., 2022) | 32.104 | 0.932 | 0.254 | 33.256 | 0.951 | 0.207 | 32.477 | 0.939 | 0.234 |
| MK-Planes* | 31.630 | 0.948 | 0.098 | 31.880 | 0.951 | 0.104 | 31.477 | 0.946 | 0.106 |
| MK-Planes | 31.677 | 0.948 | 0.098 | 32.165 | 0.952 | 0.099 | 31.751 | 0.949 | 0.099 |
| CoGS (Yu et al., 2023a) | 32.315 | 0.961 | 0.108 | 32.447 | 0.965 | 0.086 | 32.187 | 0.963 | 0.097 |
| LiveScene Qu et al. (2024) | **33.221** | 0.962 | **0.072** | 33.262 | 0.965 | 0.072 | 33.158 | 0.962 | **0.074** |
| FreeGaussian (Ours) | 33.205 | **0.967** | 0.076 | **33.922** | **0.972** | **0.071** | **33.249** | **0.969** | 0.074 |

is slightly inferior to the SOTA method in PSNR and LPIPS, it demonstrates significant advantages in scenarios where label-free guidance is required, making it particularly relevant for tasks that necessitate extensive manual labeling.

**Metric on InterReal Dataset.** As demonstrated in Table. 3, CoGS (Yu et al., 2023a) falls short of our approach on the #medium subset and fails to converge when confronted with complex scenes featuring long camera trajectories and mass of interactive objects (#challenging), revealing the limitation of existing controllable gaussian methods in modeling real-world interactive scenarios. In contrast, FreeGaussian achieves the highest SSIM of 0.893 and the lowest LPIPS of 0.165 on the #challenging subset. On the #medium subset, FreeGaussian achieves the highest PSNR compared to the current SOTA NeRF method Qu et al. (2024), showcasing its robustness in real-world scenarios with incomplete labels and its superiority in modeling real-world large-scale interactive scenarios.

Table 3: **Quantitative results on InterReal Dataset**. Our method consistently outperforms other methods across various settings, achieving the highest SSIM scores in all scenarios.

| Method | #Medium Sets | | | #Challenging Sets | | | #Avg (all 8 Sets) | | |
|---|---|---|---|---|---|---|---|---|---|
| | PSNR↑ | SSIM↑ | LPIPS↓ | PSNR↑ | SSIM↑ | LPIPS↓ | PSNR↑ | SSIM↑ | LPIPS↓ |
| NeRF (Mildenhall et al., 2020) | 20.816 | 0.682 | 0.190 | 21.169 | 0.728 | 0.337 | 20.905 | 0.694 | 0.227 |
| InstantNGP (Müller et al., 2022) | 21.700 | 0.776 | 0.215 | 21.643 | 0.745 | 0.338 | 21.686 | 0.769 | 0.245 |
| HyperNeRF (Park et al., 2021b) | 25.283 | 0.671 | 0.467 | 25.261 | 0.713 | 0.517 | 25.277 | 0.682 | 0.480 |
| K-Planes (Fridovich-Keil et al., 2023) | 27.999 | 0.813 | 0.177 | 26.427 | 0.756 | 0.331 | 27.606 | 0.799 | 0.215 |
| CoNeRF (Kania et al., 2022) | 27.501 | 0.745 | 0.367 | 26.447 | 0.734 | 0.472 | 27.237 | 0.742 | 0.393 |
| CoGS (Yu et al., 2023a) | 30.774 | 0.913 | 0.100 | ✗ | ✗ | ✗ | **30.774** | 0.913 | 0.100 |
| LiveScene Qu et al. (2024) | 30.815 | 0.911 | **0.066** | **28.436** | 0.846 | 0.185 | 30.220 | 0.895 | **0.096** |
| FreeGaussian (Ours) | **31.310** | **0.938** | 0.074 | 28.435 | **0.893** | **0.165** | 30.489 | **0.924** | 0.099 |

**Novel View Synthesis Visulization.** Figure. 15 showcases the novel view synthesis results of FreeGaussian and other methods on the both OmniSim and dataset. The results demonstrate that in addition to rendering more detailed and accurate controlled objects, our method also preserves the texture information of the background, resulting in a more comprehensive and realistic scene reconstruction. For example, LiveScene Qu et al. (2024) and MK-Planes* Qu et al. (2024) suffer from significant residual shadows when rendering three objects simultaneously (top right). CoNeRF Kania et al. (2022), although better at modeling controllable objects, loses texture information from the desktop (bottom left). CoGS Yu et al. (2023a), the first Gaussian method in the controllable domain, exhibits turbulent performance across different scenes, sometimes producing a mask for the entire

Figure 4: **View Synthesis Visualization on OmniSim and InterReal Dataset**. We show the rendering quality of our method and SOTA methods on novel view synthesis across 3 synthetic subsets and 1 real subset. In comparison with other methods, FreeGaussian achieves more realistic and detailed rendering quality, whereas other methods suffer from ghosting artifacts.

Table 4: **Model performance across size and speed**. We show the comparison of model performance in terms of number of parameters, rendering speed, and runtime memory.

| Method | Batch size | Ray samples | FPS | Parameters (MB) | Memory (GB) |
|---|---|---|---|---|---|
| CoNeRF (Kania et al., 2022) | 1024 | 256 | 0.22 | 149.58 | 71.93 |
| MK-Planes (Fridovich-Keil et al., 2023) | 4096 | 48 | 2.07 | 154.19 | 12.48 |
| MK-Planes* (Fridovich-Keil et al., 2023) | 4096 | 48 | 0.61 | 152.35 | 11.90 |
| LiveScene (Qu et al., 2024) | 4096 | 48 | 0.62 | 144.80 | 8.24 |
| CoGS (Yu et al., 2023a) | 1 | - | **215.93** | 189.70 | 25.50 |
| FreeGaussian (Ours) | 1 | - | 123.88 | **49.84** | **5.43** |

scene. In contrast, FreeGaussian method accurately models object details without compromising environmental information.

## 4.3 EVALUATION OF EFFICIENCY

To better demonstrate the advantages of FreeGaussian, we picked #seq002 from the OmniSim for statistical modeling of the number of parameters, running memory and rendering speed. Table. 4 describes that our method achieves a rendering speed of 123.88 FPS, which is significantly faster than NeRF based methods, while maintaining a relatively low memory footprint of 5.43 GB. The number of parameters in FreeGaussian is 49.84 MB, which is smaller than 1/4 the size of CoGS. These results shows that FreeGaussian is not only efficient in terms of memory usage and rendering speed but also has a smaller model size compared to existing methods.

## 4.4 ABLATION AND ANALYSIS

In this section, we conduct ablation studies to examine the contribution of each component in FreeGaussian. To facilitate a comprehensive and convincing analysis, we select three representative subsets from the OmniSim dataset: #seq001, #seq004, and #seq0015. Table. 5 shows the results of each ablation experiment.

**Effectiveness of 3D Vector Control.** We validate the effectiveness of our spherical vector controlling ability through qualitative comparisons presented in Table. 5. Compared to FreeGaussian (w/o control), FreeGaussian demonstrates significant improvements in PSNR. This is attributed to the fact that our model represents the movement of each controllable object individually using 3D vectors, which capture the object's direction and speed of motion. In contrast, models lacking 3D vector control only model objects temporally, failing to decouple time from the object's trajectory. Consequently, our model not only enables individual object control but also achieves high rendering quality, reflecting the feasibility and effectiveness of this control paradigm.

**Quality of Dynamic Gaussian Clustering.** Gaussian clustering would impact the control ability of the model, which in turn is directly influenced by the quality of the back-projected flow map. We investigate the impact of the proportion of the flow map to the total dataset on Gaussian clustering performance. Figure. 5 reveals that small keyframe ratios lead to incomplete clustering, while a 5% ratio is sufficient for achieving better clustering results. Conversely, higher ratios result in noisy clustering, which hinders subsequent control.

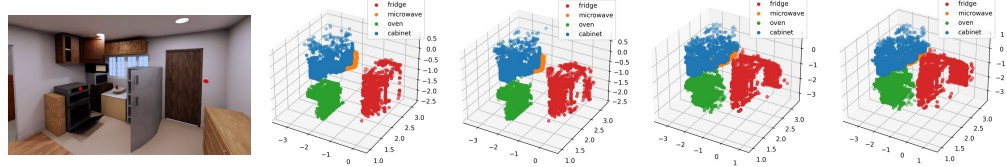

Figure 5: **Ablation study on the ratio of keyframes to total frames**. The left side presents the results for seq001. The right side illustrates the impact of varying keyframe ratios (1%, 5%, 10%, and 50%) on clustering performance.

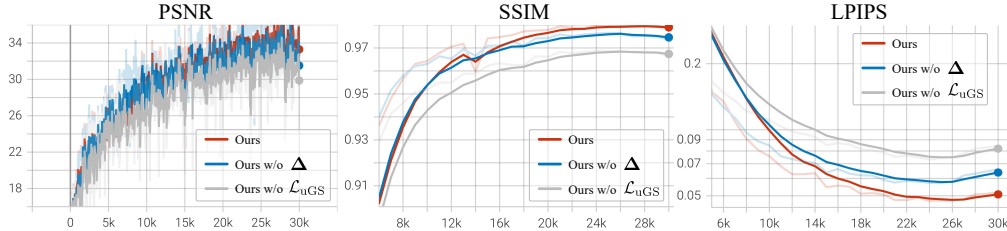

Figure 6: **Ablation study on Dynamic Gaussian Flow Loss and Flow Residual Term**. We show the training process of our model on #seq015, with training PSNR and evaluation SSIM and LPIPS.

**Flow Residual Term $\Delta$.** In *Lemma 1*, $\Delta$ is introduced to ensure the accuracy of decomposition of optical flow $\mathbf{u}$. Although this term is not exactly zero, experimental results demonstrate that it converges to zero through continuous optimization during training, shown in Figure. 6. Moreover, after convergence, this term has a negligible impact on the overall performance, as evident from the rendering metrics in Table. 5, which clearly illustrate this phenomenon.

**Dynamic Gaussian Flow Loss.** The dynamic Gaussian flow loss is designed to improve 3DGS optimization. Figure. 6 illustrates the effect of incorporating the Gaussian flow loss into our model on both convergence speed and rendering metrics. The addition of this loss term leads to a smoother and faster training process, as evident from the figure. Furthermore, the table reveals that the PSNR metrics have also improved by 1-2 dB, indicating enhanced rendering quality. This demonstrates the effectiveness of the dynamic Gaussian flow loss in faster and more effective training.

Table 5: **Ablation Study** on the subset of OmniSim Datasets. We ablate our method on 4 components in 3 seleted scenes from OmniSim Dataset and show the corresponding rendering metrics.

| Metrics | #Ablation Settings | | | | | | | |
|---|---|---|---|---|---|---|---|---|
| | | | | | Ratio of flow maps in Section. 3.2 | | | |
| | FreeGaussian in Figure. 1 | w/o 3D vector control in Section. 3.3 | w/o $\Delta$ in eq. (3) | w/o $\mathcal{L}_{uGS}$ in eq. (7) | 1% | 5% | 10% | 50% |
| PSNR | **35.31** | 33.77 | 34.24 | 33.51 | 34.61 | 35.31 | 34.61 | 31.79 |
| SSIM | **0.975** | 0.967 | 0.969 | 0.964 | 0.974 | 0.975 | 0.973 | 0.959 |
| LPIPS | **0.062** | 0.081 | 0.076 | 0.087 | 0.062 | 0.062 | 0.062 | 0.095 |

## 5 CONCLUSION AND LIMITATION

In this work, we draw the mathematical connection among optical flow, camera motion, and dynamic Gaussian flow with differential analysis, and introduce a guidance-free Gaussian Splatting method for controllable view synthesis. By leveraging the flow constraints, we refine Gaussian optimization, enabling accurate continuous Gaussian motion dynamic constraints. It not only guarantee smooth motion and improves rendering quality but also highlights interactable Gaussians and eliminates the severe dependence on manual annotations. We further introduce a 3D spherical vector controlling scheme, simplifying and accelerating interactive Gaussian modeling by bypassing the 3D Gaussian trajectory as a state representation. Extensive experiments demonstrate our superior performance in both view synthesis and scene controlling, enabling more accurate and efficient modeling of interactable content.

**Limitations:** FreeGaussian relies on optical flow estimators, and may compromise view synthesis or control robustness in lighting variation interactive environments. Future work will focus on improving the robustness of lighting variation scenes and extending the method to handle more challenging scenarios.

# FreeGaussian: Guidance-free Controllable 3D Gaussian Splats with Flow Derivatives

## Supplementary Material

### ABSTRACT

This supplementary material accompanies the main paper by providing more details for reproducibility as well as additional evaluations and qualitative results to verify the effectiveness and robustness of FreeGaussian:
▷ **Section. 6**: Dynamic Gaussian Flow Derivative Proof.
▷ **Section. 7**: Additional implementation details.
▷ **Section. 8**: Additional experimental results, including more detailed view synthesis quality comparison, clustering visualization, dual objects control capabilities and illustrations of gaussian flow map.
▷ **Section. 9**: Video demonstrations and anonymous project page: https://freegaussian.github.io.

## 6 DETAILED DYNAMIC GAUSSIAN FLOW ANALYSIS

Our insight is that dynamic Gaussian flow under instantaneous motion can be analytically decoupled from optical flow and camera motion via differential analysis with alpha composition. Considering a dynamic scene with interactive objects as shown in Figure. 2, the camera and 3D Gaussians hold separate velocities in consecutive frames 0 and $t$. Assuming a dynamic 3D Gaussian $G_i$ with velocity $\boldsymbol{v}^{\text{GS}}$, it is projected as image measurement $g_i$ under the constant camera instantaneous motion by translation velocity $\boldsymbol{v}$ and rotational velocity $\boldsymbol{\omega}$. The optical flow $\mathbf{u}$ induced by $(\boldsymbol{v}, \boldsymbol{\omega})$ of a pixel $\mathbf{m} = (x, y)^\top$ can be obtained by *Lemma 1*:

**Lemma 1:** *Dynamic Gaussian flow* $\mathbf{u}^{GS}$ *under instantaneous motion can be derived from optical flow* $\mathbf{u}$ *and camera flow* $\mathbf{u}^{Cam}$ *with the following transform eq.* (3).

$$\mathbf{u} = \mathbf{u}^{\text{Cam}} + \mathbf{u}^{\text{GS}} + \boldsymbol{\Delta},$$

$$\mathbf{u}^{\text{Cam}} = \frac{\mathbf{A}\boldsymbol{v}}{Z} + \mathbf{B}\boldsymbol{\omega}, \quad \mathbf{u}^{\text{GS}} = \mathbf{A}\sum_{i=1}^{M} T_i \alpha_i \frac{\boldsymbol{v}^{\text{GS}}}{Z_i}, \quad \boldsymbol{\Delta} = \mathbf{A}\sum_{i=1}^{M} T_i \alpha_i \boldsymbol{v}(\frac{1}{Z_i} - \frac{1}{Z}),$$

$$\mathbf{A} = \begin{bmatrix} -f_x & 0 & x - c_x \\ 0 & -f_y & y - c_y \end{bmatrix}, \quad \mathbf{B} = \begin{bmatrix} \frac{(x-c_x)(y-c_y)}{f_y} & -f_x - \frac{(x-c_x)^2}{f_x} & \frac{(y-c_y)f_x}{f_y} \\ f_y + \frac{(y-c_y)^2}{f_y} & -\frac{(x-c_x)(y-c_y)}{f_x} & -\frac{(x-c_x)f_y}{f_x} \end{bmatrix},$$

(9)

where $f_x, f_y, c_x, c_y$ are camera intrinsics, $M$ denotes the number of Gaussian projections sorted with Gaussian depth $Z_i$ intersecting the pixel $\mathbf{m}$. Flow residual term $\boldsymbol{\Delta}$ are preserved to guarantee accuracy, even when they approach zero after refined optimization.

*Proof.* We first derive the formula for 3D Gaussians derivative induced by camera rotation $\mathbf{R}(t)$, translation $\mathbf{T}(t)$, and Gaussian translation $\mathbf{T}^{\text{GS}}(t)$, which transform the 3D Gaussian $G_i$ under constant instantaneous-motion as time $t$ increasing. The equation transforming Gaussian $G_i$ from time $t$ to 0 can be formulated as:

$$\mathbf{X}_i(0) - \mathbf{T}_i^{\text{GS}}(t) = \mathbf{R}(t)\mathbf{X}_i(t) + \mathbf{T}(t), \tag{10}$$

By derivative in both sides, we reformulate the Gaussian transform in eq. (10) as:

$$-\dot{\mathbf{T}}_i^{\text{GS}}(t) = \dot{\mathbf{R}}(t)\mathbf{X}_i(t) + \mathbf{R}(t)\dot{\mathbf{X}}_i(t) + \dot{\mathbf{T}}(t), \tag{11}$$

$$\dot{\mathbf{X}}_i(t) = -\mathbf{R}^\top(t)\dot{\mathbf{R}}(t)\mathbf{X}_i(t) - \mathbf{R}^\top(t)\dot{\mathbf{T}}(t) - \mathbf{R}^\top(t)\dot{\mathbf{T}}_i^{\text{GS}}(t). \tag{12}$$

According to Possion's equation (Ling et al.; Heeger & Jepson, 1992), the rotation and translation velocities can be defined with $\mathbf{R}^\top(t)\dot{\mathbf{R}}(t) = [\boldsymbol{\omega}]_\times$, $\mathbf{R}^\top(t)\dot{\mathbf{T}}(t) = \boldsymbol{v}$ and $\mathbf{R}^\top(t)\dot{\mathbf{T}}^{\text{GS}}(t) = \boldsymbol{v}^{\text{GS}}$. By

substituting the above equations into eq. (12) and omitting the time notation, we obtain the simplicity results:

$$\dot{\mathbf{X}}_i = -[\boldsymbol{\omega}]_\times \mathbf{X}_i - \boldsymbol{v} - \boldsymbol{v}^{\text{GS}}, \tag{13}$$

where $\boldsymbol{v}^{\text{GS}}$ presents the velocity of the dynamic 3D Gaussian $G_i$. Then, the camera projection model with respect to $\mathbf{X}_i$ is:

$$Z_i[\mu_i; 1] = \mathbf{K}\mathbf{X}_i. \tag{14}$$

In order to derive the dynamic Gaussian flow $\mathbf{u}_i^{\text{GS}}$ in the 2D image plane, we derivative on both sides and obtain the differential of the projected image coordinates, namely the optical flow, in relation to the projection parameters:

$$\mathbf{u}_i^{\text{GS}} = \begin{bmatrix} \frac{f_x}{Z} & 0 & -\frac{f_x X}{Z^2} \\ 0 & \frac{f_y}{Z} & -\frac{f_y Y}{Z^2} \end{bmatrix} \dot{\mathbf{X}}_i. \tag{15}$$

By substituting the above equations eq. (15) into eq. (13), we obtain the dynamic Gaussian flow decomposition $\mathbf{u}_i^{\text{GS}}$ in individual Gaussian $G_i$ as:

$$\mathbf{u}_i = \frac{\mathbf{A}\boldsymbol{v}}{Z_i} + \mathbf{B}\boldsymbol{\omega} + \frac{\mathbf{A}\boldsymbol{v}^{\text{GS}}}{Z_i} = \left(\frac{\mathbf{A}\boldsymbol{v}}{Z} + \mathbf{B}\boldsymbol{\omega}\right) + \frac{\mathbf{A}\boldsymbol{v}^{\text{GS}}}{Z_i} + \left(\frac{\mathbf{A}\boldsymbol{v}}{Z_i} - \frac{\mathbf{A}\boldsymbol{v}}{Z}\right) \tag{16}$$

With alpha composition, we weight the flow with $w_i = \frac{T_i \alpha_i}{\Sigma_i T_i \alpha_i}$ in both sides and proof the mathematical relation described in eq. (3). $\qquad\square$

# 7 ADDITIONAL IMPLEMENTATION DETAILS

**Implementation Details.** FreeGaussian is implemented based on nerfstudio (Tancik et al., 2023) and gsplat (Ye et al., 2024). We use RAFT Teed & Deng (2020); Contributors (2021) for optical flow prediction and perform DBSCAN clustering from dynamic Gaussian flow with Euclidean metric, $\epsilon = 0.05$ and minimal samples = 5. The cluster center corresponding to each Gaussian is encoded with hash grids and decoded with an 8-layer MLP with 256 neurons. The model is trained on an NVIDIA GeForce RTX 4090 GPU for 60k steps, using Adam optimizer with learning rate $1.6e^{-4}$ and batch size 1. The coarse-to-fine training process lasts 30 minutes and is divided into 3 stages, including 500 steps of canonical warmup, 30k steps 4d deformable training, and 30k steps of full training. For all experiments, we set loss weights of $\mathcal{L}_{\text{RGB}}$, $\mathcal{L}_{\text{D-SSIM}}$, and $\mathcal{L}_{\text{uGS}}$ as $\lambda = 0.8$, $(1-\lambda) = 0.2$, and $\beta = 0.5$, respectively.

**The CUDA implementation** of the proposed Dynamic Gaussian Flow Constrain is based on gsplat (Ye et al., 2024). We implemented a minimal modification to the source code, creating a mapping from pixel coordinates to Gaussian sphere identifiers and their associated weights. Due to the potential intersection of pixel coordinates with numerous Gaussian splats, we opted to store the top 50 Gaussian ellipsoid indices per pixel and perform reweighting with $w_i = \frac{T_i \alpha_i}{\Sigma_i^{50} T_i \alpha_i}$ as necessary. Backpropagation only updates the gradients of associated weights, not the pixel coordinates to Gaussian mapping.

**Dynamic Gaussian Clustering.** Gaussian clustering would impact the control ability of the model, which in turn is directly influenced by the quality of the back-projected flow map. We configure the frame interval to be 1 and establish correspondences between the optical flows of adjacent frames. By leveraging eq. (3), we compute the Gaussian interaction flow. Next, by randomly sampling 5% of the interaction flow map as keyframes, we perform back-projection and apply DBSCAN clustering to obtain dynamic Gaussians. Small keyframe ratios lead to incomplete clustering, while a 5% ratio is sufficient for achieving better clustering results. Conversely, higher ratios result in noisy clustering, which hinders subsequent control.

**Algorithm Implementation.** Algorithm 1 provided detailed implementation pseudo code of FreeGaussian, including the deformable 3D Gaussian pre-training, dynamic Gaussian flow decouple, DBSCAN clustering, and Self-guide control with dynamic 3D Gaussian.

# 8 ADDITIONAL EXPERIMENTAL RESULTS

**View Synthesis Quality Comparison on OmniSim and InterReal dataset** We present detailed quantitative results on the OmniSim and InterReal datasets in Table. 6 and Table. 7, respectively.

---

**Algorithm 1:** Controllable 3D Gaussian Splats with Flow Derivatives

---

**Input** : Set camera stream $\{\mathbf{P}(t), \mathbf{I}(t)\}$ and initialize 3D Gaussians $\mathbf{G}^0$.

**Output:** Controllable 3D Gaussians $\mathbf{G}^*$ with Network $\Theta^*$.

1 ▷ pre-train a deformable 3DGS $\mathbf{G}'$;

2 ▽ Dynamic Gaussian Flow Decouple;

3 **for** *Each continuous camera views* $\mathbf{P}(0), \mathbf{P}(t)$ **do**

4     Estimate optical flow $\mathbf{u}$ and caculate camera flow $\mathbf{u}^{\text{Cam}}$ using eq. (3);

5     Calculate dynamic gaussian flow $\mathbf{u}^{\text{GS}}$ using eq. (4);

6     Back project binarized dynamic Gaussian flow $\mathbf{bin}(\mathbf{u}^{\text{GS}})$ to 3DGS: $g_i \rightarrow \mathcal{D}$;

7 **end**

8 ▷ **DBSCAN** clustering and caculate trajectory $\varsigma(t, k)$;

9 ▽ Self-guided Control with Dynamic 3DGS;

10 **while** *(not reach max iteration) and (not satisfy stopping criteria)* **do**

11     **for** *Each continuous pair* $< \mathbf{P}(t), \mathbf{I}(t) >$ **do**

12        Encode coordinates $\mathbf{v}_c^i = \varsigma(t, k) - \varsigma(0, k)$ with hash grid: $\mathbf{E}(\mathbf{v}_c^i)$;

13        Forward pass and rasterize with $\mathbf{G}^*$ and $\mathbf{E}(\varsigma)$: $\mathbf{I}, \mathbf{u}^{\text{GS}} = \Theta(\mathbf{G}^*, \mathbf{E}(\varsigma))$;

14        Calculate loss $\mathcal{L}_{\text{uGS}}, \mathcal{L}_{\text{RGB}}, \mathcal{L}_{\text{D-SSIM}}$ using eq. (4) and optimize with Gradient Descent;

15        Update $\Theta^*$ and $\mathbf{G}^*$;

16     **end**

17 **end**

18 ▽ Controlling with FreeGaussian;

19 **for** *Each control camera view and 3d vector* $\mathbf{v}_c'$ **do**

20     Back-project to query Gaussian $G_i$ ;

21     Perform hash encoding: $\mathbf{E}(\mathbf{v}_c')$;

22     Forward pass $\Theta^*$ and rasterize with $\boldsymbol{f}_{\Theta^*}(\mathbf{X}_i, \mathbf{v}_c')$

23 **end**

---

Our method demonstrates significant advantages on both the #easy and #medium subsets of the OmniSim dataset. Additionally, it achieves notable scores on the #medium subset of the InterReal dataset. A multitude of metrics indicate that our model excels in rendering on both simulated and real datasets, underscoring its superiority. While the metric improvements may be modest compared to current SOTA NeRF methods, our approach offers a substantial advantage by introducing a novel guidance-free training paradigm that significantly reduces the label requirements, thereby enhancing its real-world applicability. We report scores as NaN if the model fails to converge or runs out of memory during training multiple times.

**More Detailed Rendering Comparison** We show additional visual comparisons in Figure. 7, Figure. 8, showcasing our method's superior performance on the OmniSim and InterReal datasets. Our approach excels in reconstructing detailed and accurate object representations. Notably, our method generates more accurate object shapes and background textures compared to existing approaches.

**More Detailed on Individual Control Capability** We present two examples, as shown in Figure. 9 and Figure. 10, to demonstrate the model's capability to control individual objects. Our method offers superior control over the objects within the scene, enabling the model to implement attribute combinations that were never seen in the training data.

**More Detailed Clustering Visualization** Figure. 11 illustrates the clustering results of our method across various scenarios. As demonstrated, the majority of Gaussian clusters are accurately grouped around controllable entities, particularly in relation to the moving components. This can be attributed to the successful decoupling of the interaction flow, a feature that enables the Gaussian clusters to concentrate more effectively on the motion rendering.

**More illustrations of dynamic Gaussian flow map** We provide a more detailed visualization of highlighting dynamic Gaussian capabilities in Figure. 12. The experimental results show that, despite the presence of complex camera motion and interactive body motion, the proposed approach successfully decouples the Gaussian dynamics, producing accurate and detailed flow maps. Notably, objects exhibiting complex topological structure changes, such as boxes or dishwashers, can be ef-

Table 6: **Detailed Quantitative Results on OmniSim Dataset**. FreeGaussian outperforms prior works on most metrics, especially the #easy and #medium subsets.

| Dataset | Metric | NeRF | Instant-NGP | HyperNeRF | CoNeRF | K-Planes | MK-Planes | MK-Planes* | LiveScene | CoGS | FreeGaussian |
|---|---|---|---|---|---|---|---|---|---|---|---|
| seq001_Rs_int | psnr | 25.941 | 25.768 | NaN | 34.035 | 33.136 | 32.169 | 32.092 | 34.784 | 32.211 | 36.335 |
| seq001_Rs_int | ssim | 0.931 | 0.933 | NaN | 0.957 | 0.953 | 0.946 | 0.946 | 0.974 | 0.968 | 0.980 |
| seq001_Rs_int | lpips | 0.118 | 0.113 | NaN | 0.135 | 0.093 | 0.110 | 0.110 | 0.048 | 0.068 | 0.046 |
| seq002_Rs_int | psnr | 28.616 | 28.660 | NaN | 34.286 | 34.765 | 36.532 | 34.580 | 35.190 | 34.497 | 34.979 |
| seq002_Rs_int | ssim | 0.950 | 0.946 | NaN | 0.951 | 0.967 | 0.976 | 0.968 | 0.969 | 0.979 | 0.976 |
| seq002_Rs_int | lpips | 0.096 | 0.112 | NaN | 0.217 | 0.074 | 0.036 | 0.074 | 0.070 | 0.051 | 0.060 |
| seq003_Ihlen_1_int | psnr | 26.720 | 28.255 | 33.551 | 34.700 | 35.217 | 34.758 | 34.753 | 35.323 | 36.816 | 36.094 |
| seq003_Ihlen_1_int | ssim | 0.940 | 0.944 | 0.946 | 0.953 | 0.964 | 0.966 | 0.966 | 0.966 | 0.980 | 0.974 |
| seq003_Ihlen_1_int | lpips | 0.120 | 0.121 | 0.268 | 0.244 | 0.097 | 0.087 | 0.090 | 0.094 | 0.077 | 0.077 |
| seq004_Ihlen_1_int | psnr | 30.847 | 31.800 | 31.115 | 32.684 | 36.157 | 34.863 | 35.000 | 36.712 | 31.055 | 35.700 |
| seq004_Ihlen_1_int | ssim | 0.927 | 0.942 | 0.878 | 0.888 | 0.955 | 0.919 | 0.926 | 0.962 | 0.915 | 0.965 |
| seq004_Ihlen_1_int | lpips | 0.104 | 0.102 | 0.389 | 0.366 | 0.085 | 0.145 | 0.135 | 0.072 | 0.209 | 0.086 |
| seq005_Beechwood_0_int | psnr | 27.183 | 27.295 | 30.699 | 32.549 | 31.944 | 33.195 | 33.098 | 33.623 | 33.664 | 33.778 |
| seq005_Beechwood_0_int | ssim | 0.930 | 0.937 | 0.906 | 0.927 | 0.944 | 0.961 | 0.959 | 0.962 | 0.978 | 0.973 |
| seq005_Beechwood_0_int | lpips | 0.127 | 0.112 | 0.291 | 0.245 | 0.105 | 0.076 | 0.080 | 0.072 | 0.058 | 0.063 |
| seq006_Beechwood_0_int | psnr | 27.988 | 28.150 | 29.513 | 30.058 | 31.861 | 31.541 | 31.521 | 32.206 | 31.272 | 32.067 |
| seq006_Beechwood_0_int | ssim | 0.938 | 0.938 | 0.907 | 0.917 | 0.951 | 0.951 | 0.951 | 0.959 | 0.974 | 0.971 |
| seq006_Beechwood_0_int | lpips | 0.103 | 0.119 | 0.314 | 0.283 | 0.097 | 0.095 | 0.096 | 0.077 | 0.059 | 0.058 |
| seq007_Beechwood_0_int | psnr | 23.201 | 22.902 | 31.259 | 33.451 | 30.979 | 30.136 | 30.089 | 30.360 | 27.367 | 33.748 |
| seq007_Beechwood_0_int | ssim | 0.885 | 0.886 | 0.913 | 0.935 | 0.938 | 0.942 | 0.942 | 0.946 | 0.893 | 0.969 |
| seq007_Beechwood_0_int | lpips | 0.220 | 0.219 | 0.289 | 0.229 | 0.140 | 0.120 | 0.121 | 0.107 | 0.219 | 0.084 |
| seq008_Benevolence_1_int | psnr | 25.750 | 25.574 | 32.691 | 34.319 | 31.914 | 30.926 | 30.916 | 33.393 | 33.795 | 33.855 |
| seq008_Benevolence_1_int | ssim | 0.943 | 0.940 | 0.945 | 0.960 | 0.948 | 0.941 | 0.941 | 0.970 | 0.980 | 0.975 |
| seq008_Benevolence_1_int | lpips | 0.113 | 0.123 | 0.229 | 0.185 | 0.107 | 0.118 | 0.116 | 0.067 | 0.072 | 0.068 |
| seq009_Benevolence_1_int | psnr | 24.326 | 24.386 | 29.596 | 31.225 | 32.836 | 31.500 | 31.471 | 32.030 | 33.205 | 31.960 |
| seq009_Benevolence_1_int | ssim | 0.921 | 0.922 | 0.897 | 0.932 | 0.956 | 0.954 | 0.953 | 0.962 | 0.975 | 0.959 |
| seq009_Benevolence_1_int | lpips | 0.124 | 0.128 | 0.327 | 0.248 | 0.090 | 0.088 | 0.090 | 0.071 | 0.074 | 0.089 |
| seq010_Merom_1_int | psnr | 22.927 | 22.765 | 28.985 | 31.092 | 30.120 | 29.461 | 29.396 | 30.029 | 30.254 | 30.622 |
| seq010_Merom_1_int | ssim | 0.917 | 0.925 | 0.939 | 0.957 | 0.960 | 0.960 | 0.959 | 0.966 | 0.974 | 0.971 |
| seq010_Merom_1_int | lpips | 0.173 | 0.158 | 0.275 | 0.233 | 0.093 | 0.087 | 0.088 | 0.074 | 0.065 | 0.080 |
| seq011_Merom_1_int | psnr | 26.732 | 27.077 | NaN | 30.483 | 33.394 | 32.951 | 32.910 | 33.426 | 31.767 | 33.014 |
| seq011_Merom_1_int | ssim | 0.932 | 0.933 | NaN | 0.932 | 0.959 | 0.959 | 0.959 | 0.960 | 0.968 | 0.966 |
| seq011_Merom_1_int | lpips | 0.112 | 0.117 | NaN | 0.246 | 0.074 | 0.073 | 0.072 | 0.068 | 0.091 | 0.079 |
| seq012_Pomaria_1_int | psnr | 26.856 | 27.074 | NaN | 33.065 | 35.185 | 32.248 | 32.209 | 33.367 | 37.284 | 34.104 |
| seq012_Pomaria_1_int | ssim | 0.936 | 0.943 | NaN | 0.954 | 0.972 | 0.966 | 0.966 | 0.969 | 0.985 | 0.972 |
| seq012_Pomaria_1_int | lpips | 0.138 | 0.126 | NaN | 0.199 | 0.059 | 0.075 | 0.075 | 0.061 | 0.047 | 0.067 |
| seq013_Pomaria_1_int | psnr | 25.277 | 24.018 | NaN | 33.682 | 30.860 | 30.390 | 30.299 | 33.592 | 32.868 | 32.730 |
| seq013_Pomaria_1_int | ssim | 0.925 | 0.930 | NaN | 0.964 | 0.943 | 0.931 | 0.930 | 0.970 | 0.981 | 0.970 |
| seq013_Pomaria_1_int | lpips | 0.154 | 0.161 | NaN | 0.166 | 0.123 | 0.162 | 0.164 | 0.056 | 0.045 | 0.072 |
| seq014_Wainscott_0_int | psnr | 26.011 | 25.966 | NaN | 29.580 | 32.517 | 30.511 | 30.504 | 31.197 | 31.885 | 31.709 |
| seq014_Wainscott_0_int | ssim | 0.927 | 0.924 | NaN | 0.925 | 0.955 | 0.951 | 0.951 | 0.952 | 0.969 | 0.958 |
| seq014_Wainscott_0_int | lpips | 0.105 | 0.116 | NaN | 0.244 | 0.077 | 0.082 | 0.083 | 0.083 | 0.067 | 0.084 |
| seq015_Wainscott_0_int | psnr | 27.257 | 27.191 | NaN | 32.307 | 30.721 | 28.288 | 28.134 | 34.266 | 32.949 | 35.014 |
| seq015_Wainscott_0_int | ssim | 0.953 | 0.951 | NaN | 0.962 | 0.955 | 0.942 | 0.942 | 0.976 | 0.975 | 0.980 |
| seq015_Wainscott_0_int | lpips | 0.080 | 0.092 | NaN | 0.202 | 0.083 | 0.110 | 0.108 | 0.050 | 0.078 | 0.047 |
| seq016_Wainscott_0_int | psnr | 21.953 | 21.660 | 28.364 | 30.205 | 30.414 | 28.915 | 28.710 | 29.746 | 31.965 | 31.096 |
| seq016_Wainscott_0_int | ssim | 0.897 | 0.895 | 0.909 | 0.935 | 0.951 | 0.952 | 0.951 | 0.955 | 0.976 | 0.967 |
| seq016_Wainscott_0_int | lpips | 0.175 | 0.194 | 0.327 | 0.260 | 0.089 | 0.086 | 0.087 | 0.083 | 0.066 | 0.075 |
| seq017_Benevolence_1_int | psnr | 26.364 | 26.367 | 27.533 | 30.349 | 29.833 | 29.254 | 26.565 | 31.645 | 28.701 | 28.347 |
| seq017_Benevolence_1_int | ssim | 0.927 | 0.920 | 0.897 | 0.923 | 0.937 | 0.933 | 0.887 | 0.948 | 0.970 | 0.958 |
| seq017_Benevolence_1_int | lpips | 0.128 | 0.143 | 0.318 | 0.238 | 0.118 | 0.119 | 0.218 | 0.093 | 0.073 | 0.089 |
| seq018_Benevolence_1_int | psnr | 28.236 | 24.296 | 32.551 | 34.297 | 34.690 | 33.049 | 33.002 | 34.187 | 34.963 | 33.659 |
| seq018_Benevolence_1_int | ssim | 0.918 | 0.809 | 0.911 | 0.936 | 0.951 | 0.953 | 0.952 | 0.958 | 0.976 | 0.966 |
| seq018_Benevolence_1_int | lpips | 0.145 | 0.342 | 0.293 | 0.248 | 0.093 | 0.090 | 0.091 | 0.081 | 0.114 | 0.085 |
| seq019_Rs_int | psnr | 20.059 | 20.854 | 33.119 | 34.598 | 34.462 | 33.679 | 33.653 | 35.223 | 25.947 | 34.097 |
| seq019_Rs_int | ssim | 0.794 | 0.808 | 0.950 | 0.963 | 0.956 | 0.963 | 0.962 | 0.969 | 0.879 | 0.970 |
| seq019_Rs_int | lpips | 0.425 | 0.424 | 0.270 | 0.225 | 0.106 | 0.087 | 0.089 | 0.068 | 0.327 | 0.089 |
| seq020_Merom_1_int | psnr | 23.273 | 24.074 | 31.280 | 32.580 | 30.462 | 30.655 | 30.626 | 32.869 | 31.280 | 32.068 |
| seq020_Merom_1_int | ssim | 0.823 | 0.852 | 0.970 | 0.914 | 0.929 | 0.919 | 0.918 | 0.954 | 0.970 | 0.954 |
| seq020_Merom_1_int | lpips | 0.306 | 0.259 | 0.086 | 0.276 | 0.140 | 0.139 | 0.142 | 0.078 | 0.086 | 0.095 |

fectively isolated. This outcome substantiates the efficacy and unsupervised exploration capabilities of the proposed method for interactive Gaussian discovery.

# 9 VIDEOS DEMONSTRATION AND ANONYMOUS LINK

We provide a video of our proposed method FreeGaussian along with this document to demonstrate the interactive scene reconstruction and multimodal control capabilities. Please refer to the anonymous link: https://freegaussian.github.io for more information.

Table 7: **Detailed Quantitative Results on InterReal Dataset**. FreeGaussian consistently outperforms all other methods in most sequences. Across most sequences, FreeGaussian maintains high PSNR and SSIM, with low LPIPS, indicating that it excels in both numerical image quality and perceptual similarity.

| Dataset | Metric | NeRF | Instant-NGP | HyperNeRF | CoNeRF | K-Planes | LiveScene | CoGS | FreeGaussian |
|---|---|---|---|---|---|---|---|---|---|
| seq001_transformer | psnr | 20.094 | 20.619 | 24.651 | 27.260 | 26.881 | 30.396 | 31.067 | 31.067 |
| seq001_transformer | ssim | 0.725 | 0.805 | 0.638 | 0.739 | 0.791 | 0.912 | 0.943 | 0.943 |
| seq001_transformer | lpips | 0.182 | 0.167 | 0.495 | 0.355 | 0.185 | 0.060 | 0.060 | 0.060 |
| seq002_transformer | psnr | 20.093 | 20.028 | 24.433 | 26.917 | 26.232 | 29.706 | 30.513 | 30.513 |
| seq002_transformer | ssim | 0.736 | 0.778 | 0.635 | 0.732 | 0.763 | 0.899 | 0.938 | 0.938 |
| seq002_transformer | lpips | 0.210 | 0.196 | 0.477 | 0.357 | 0.223 | 0.069 | 0.062 | 0.062 |
| seq003_door | psnr | 20.001 | 20.652 | 27.144 | 29.850 | 29.278 | 32.709 | 31.998 | 31.998 |
| seq003_door | ssim | 0.785 | 0.831 | 0.878 | 0.922 | 0.920 | 0.960 | 0.962 | 0.962 |
| seq003_door | lpips | 0.250 | 0.250 | 0.316 | 0.231 | 0.101 | 0.044 | 0.071 | 0.071 |
| seq004_dog | psnr | 20.044 | 20.206 | 25.691 | 28.567 | 30.350 | 32.519 | 32.455 | 33.555 |
| seq004_dog | ssim | 0.723 | 0.819 | 0.730 | 0.815 | 0.894 | 0.943 | 0.950 | 0.960 |
| seq004_dog | lpips | 0.196 | 0.178 | 0.435 | 0.324 | 0.107 | 0.049 | 0.074 | 0.063 |
| seq005_sit | psnr | 21.558 | 24.211 | 24.944 | 26.252 | 27.970 | 30.161 | 27.169 | 30.236 |
| seq005_sit | ssim | 0.480 | 0.727 | 0.573 | 0.633 | 0.773 | 0.886 | 0.767 | 0.912 |
| seq005_sit | lpips | 0.178 | 0.236 | 0.543 | 0.463 | 0.207 | 0.084 | 0.232 | 0.098 |
| seq006_stand | psnr | 23.109 | 24.483 | 24.833 | 26.159 | 27.285 | 29.400 | 31.442 | 30.489 |
| seq006_stand | ssim | 0.643 | 0.699 | 0.574 | 0.627 | 0.736 | 0.868 | 0.919 | 0.913 |
| seq006_stand | lpips | 0.123 | 0.260 | 0.538 | 0.470 | 0.237 | 0.089 | 0.104 | 0.092 |
| seq007_flower | psnr | 21.150 | 21.813 | 25.334 | 26.854 | 26.545 | 28.208 | 28.435 | 28.435 |
| seq007_flower | ssim | 0.721 | 0.747 | 0.712 | 0.748 | 0.759 | 0.844 | 0.893 | 0.893 |
| seq007_flower | lpips | 0.302 | 0.319 | 0.489 | 0.425 | 0.321 | 0.188 | 0.165 | 0.165 |
| seq008_office | psnr | 21.187 | 21.474 | 25.188 | 26.040 | 26.309 | 28.663 | 27.510 | 27.620 |
| seq008_office | ssim | 0.735 | 0.743 | 0.714 | 0.720 | 0.754 | 0.848 | 0.897 | 0.872 |
| seq008_office | lpips | 0.371 | 0.358 | 0.545 | 0.520 | 0.341 | 0.181 | 0.138 | 0.181 |

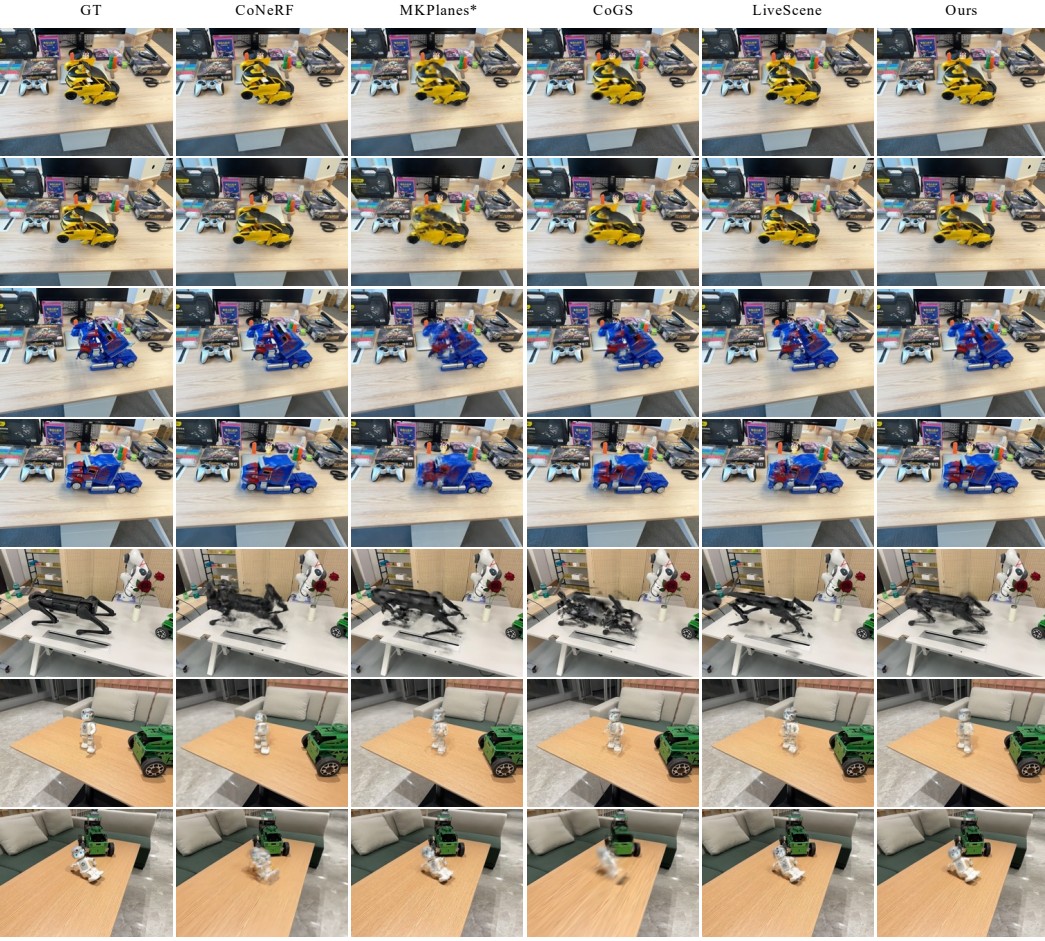

Figure 7: **View Synthesis Visualization on InterReal Dataset**. We compare our method with SOTA methods on RGB rendering across real scenes. FreeGaussian obtained more detailed and accurate representations of the objects. While other methods fail to capture the object's shape and cause significant artifacts.

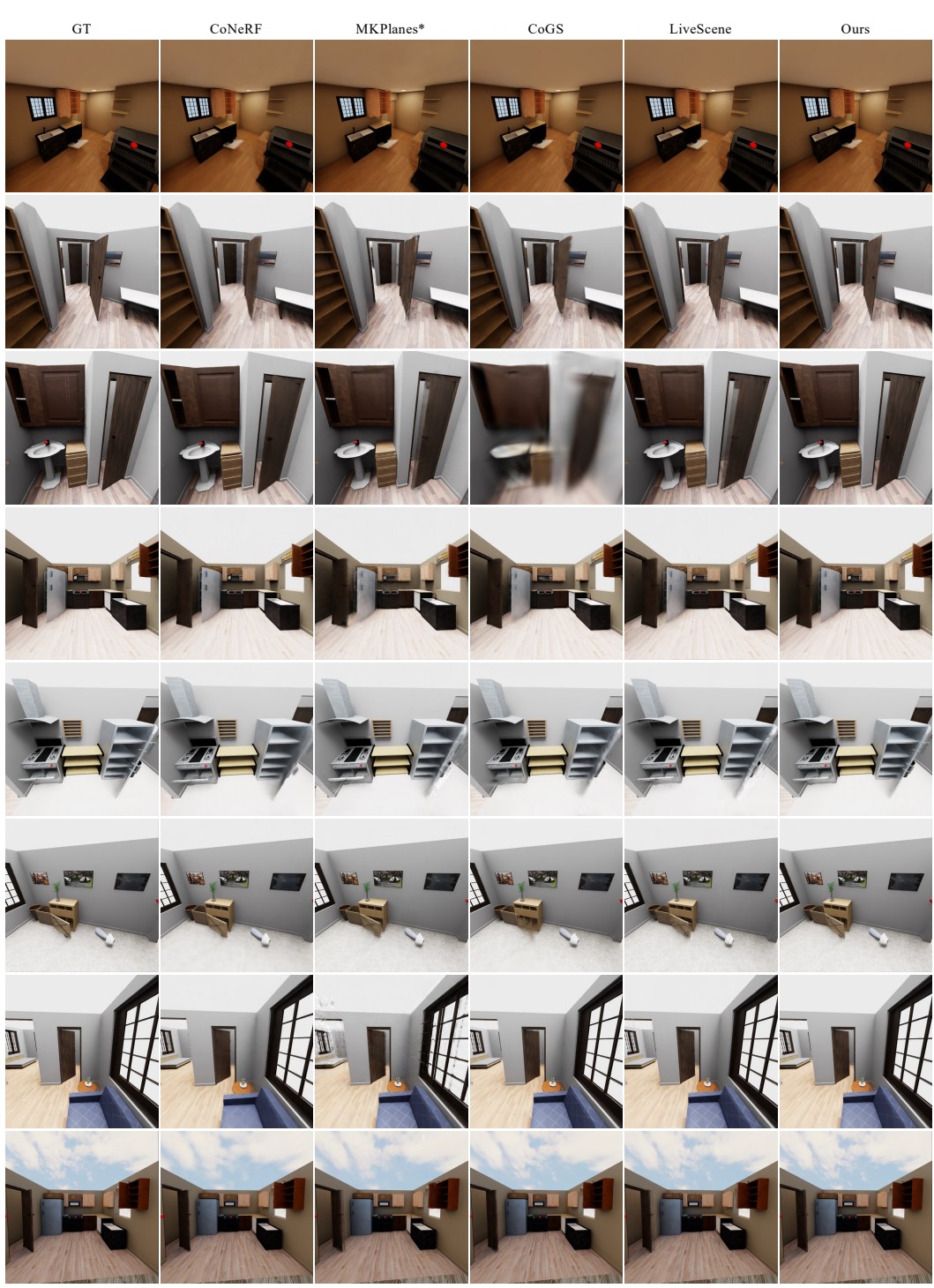

Figure 8: **View Synthesis Visualization on OmniSim Dataset**. Compared with the other methods, FreeGaussian reconstructs clear and accurate object shapes and textures.

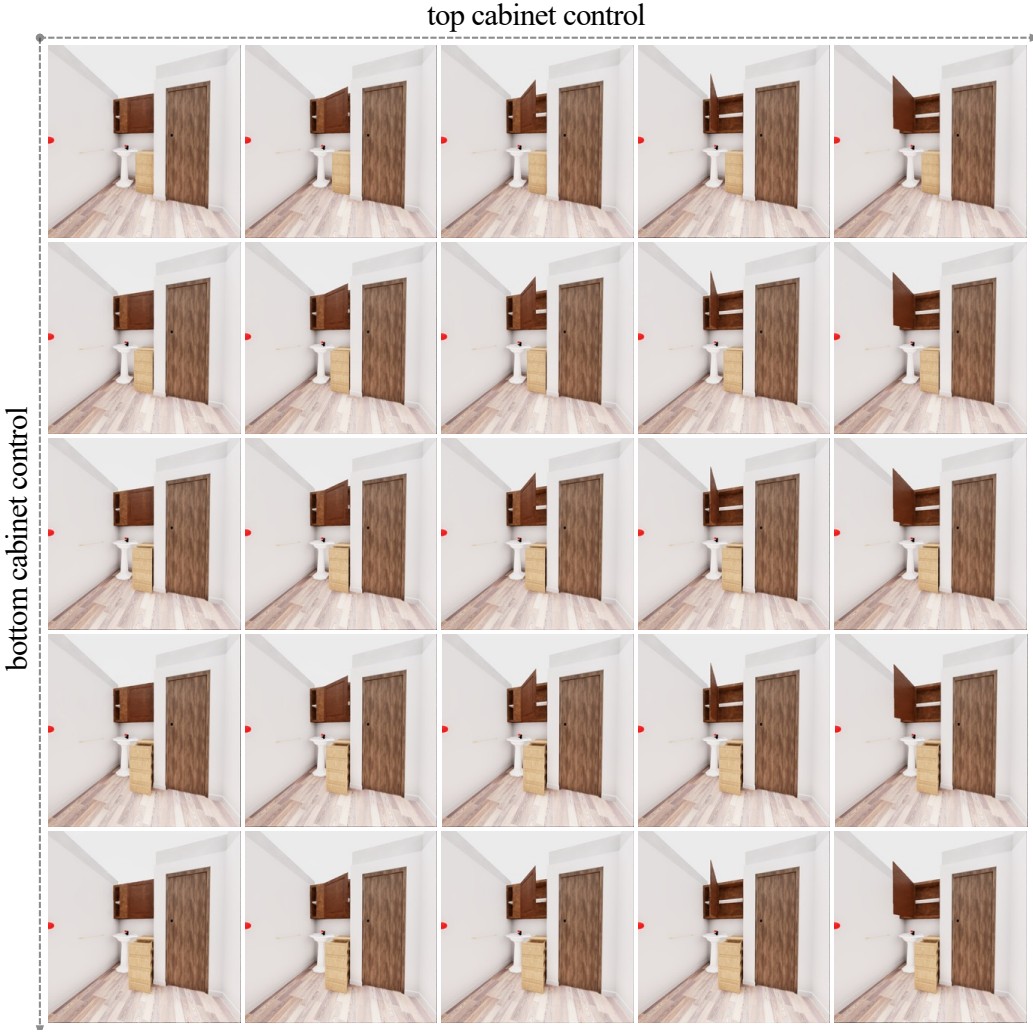

Figure 9: **Visualization of Dual Attributes Combination on #seq007**. The model allows for independent control of both the upper and lower cabinets, highlighting its superior capability for precise, individual control.

Figure 10: **Visualization of Dual Attributes Combination on #seq009**. The model enables independent control of the chest and the door, showcasing its advanced ability for precise, individual operation.

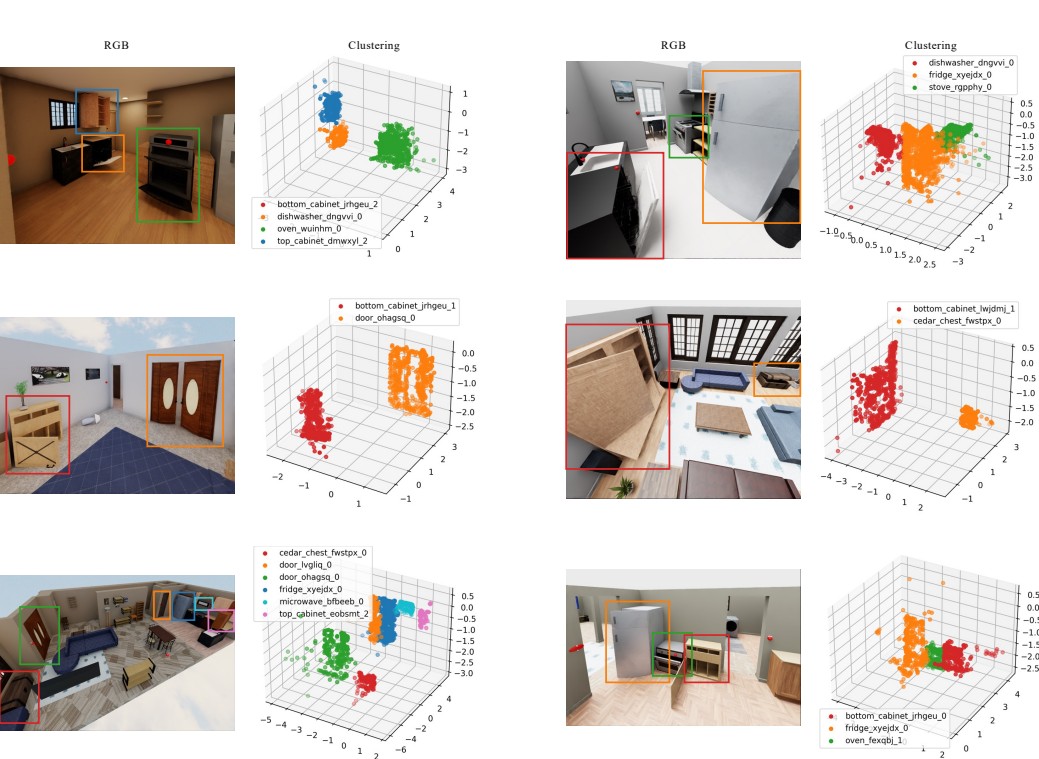

Figure 11: **Visualization of DBSCAN clustering**. After successfully training the 4D Gaussian field, we apply DBSCAN and the interaction flow to identify the key Gaussian spheres corresponding to the controllable objects.

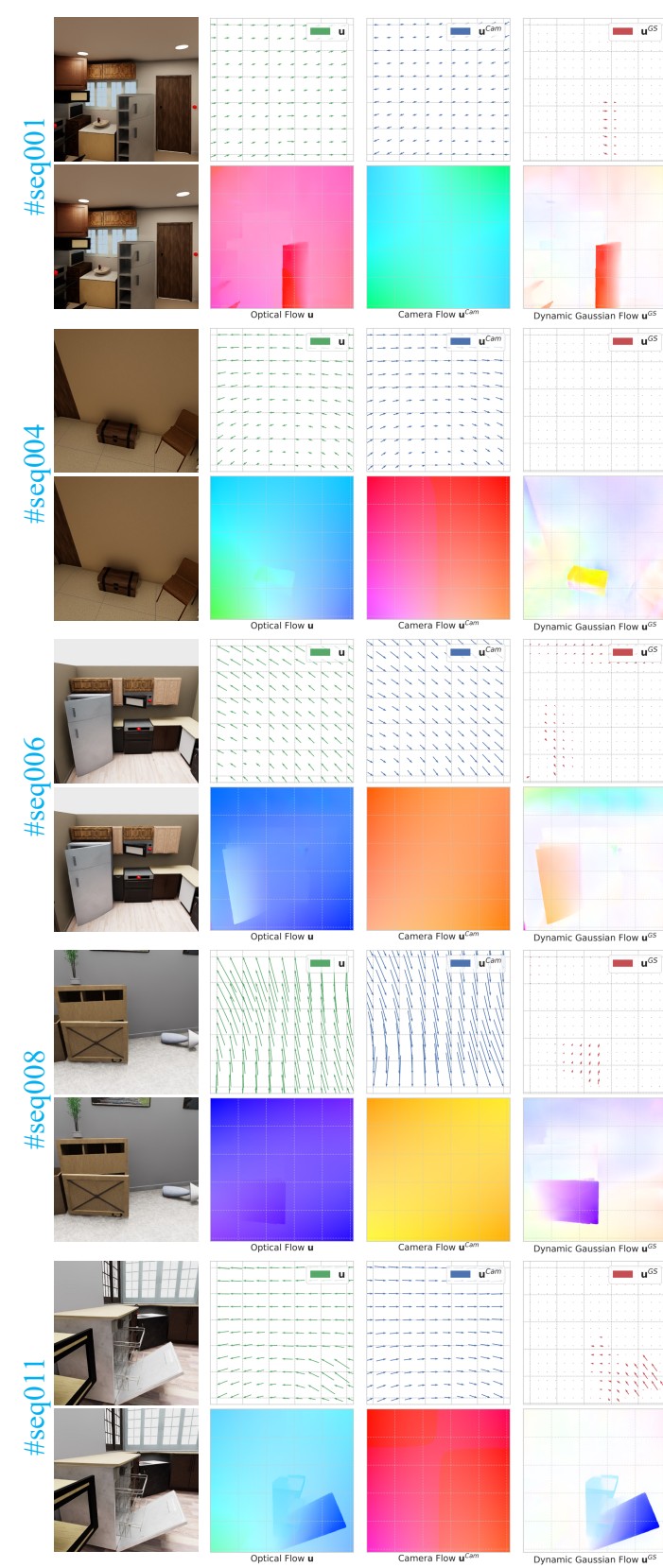

Figure 12: More illustrations of dynamic Gaussian flow map under dynamic scenes of OmniSim. For dynamic scenes with interactive objects and complex camera motions (translation and rotation), the dynamic Gaussian flow map will highlight interactive 3D Gaussians, and demonstrate the effectiveness of proposed Dynamic Gaussian Flow derivatives in eq. (3).

GT HyperNeRF CoNeRF CoGS LiveScene Ours

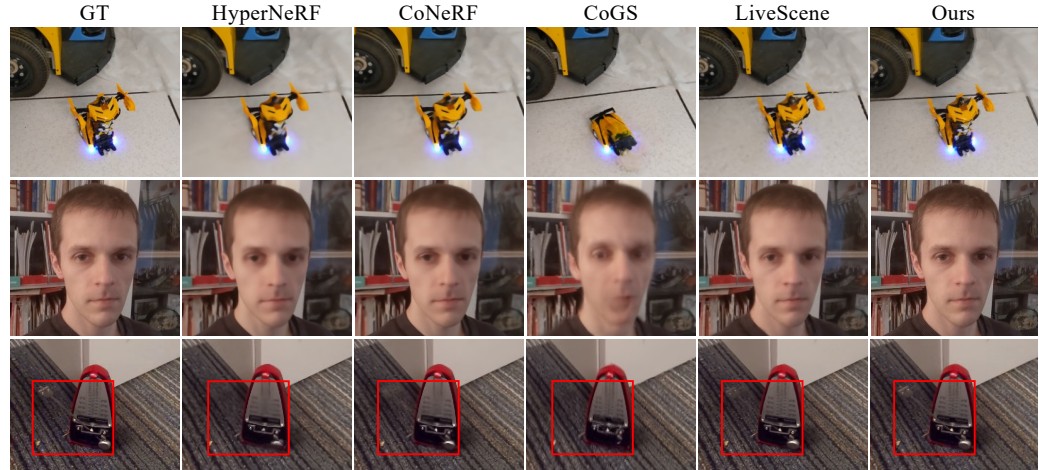

Figure 13: View Synthesis Visualization on CoNeRF Controllable Dataset.

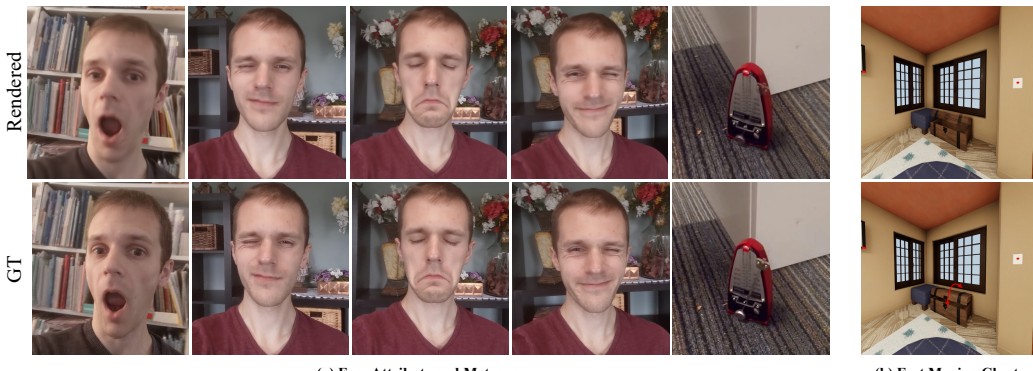

(a) Face Attribute and Metronome  (b) Fast Moving Chest

Figure 14: (a) Qualitative results on CoNeRF Faces and Metronome subsets. (b) Qualitative result on failure case of fast moving chest.

Cook Spinach  Cut Roasted Beef  Coffee Martini  Sear Steak  Flame Steak  Flame Salmon

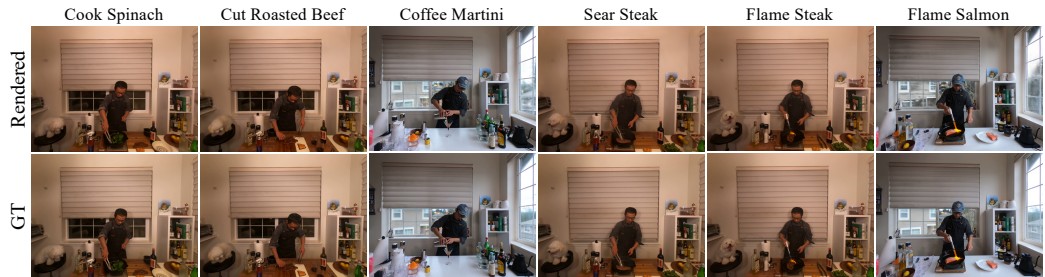

Figure 15: View Synthesis Visualization on DyNeRF Dataset.

Table 8: Quantitative results between GaussianFlow and FreeGaussian.

|       |              | #seq001 | #seq004 | #seq015 | Mean  |
|-------|--------------|---------|---------|---------|-------|
| PSNR  | GaussianFlow | 32.21   | 34.64   | 31.36   | 32.74 |
|       | Ours         | **36.34** | **35.70** | **35.01** | **35.68** |
| SSIM  | GaussianFlow | 0.968   | 0.958   | 0.966   | 0.964 |
|       | Ours         | **0.980** | **0.965** | **0.980** | **0.975** |
| LPIPS | GaussianFlow | 0.068   | 0.103   | 0.084   | 0.085 |
|       | Ours         | **0.046** | **0.086** | **0.047** | **0.060** |

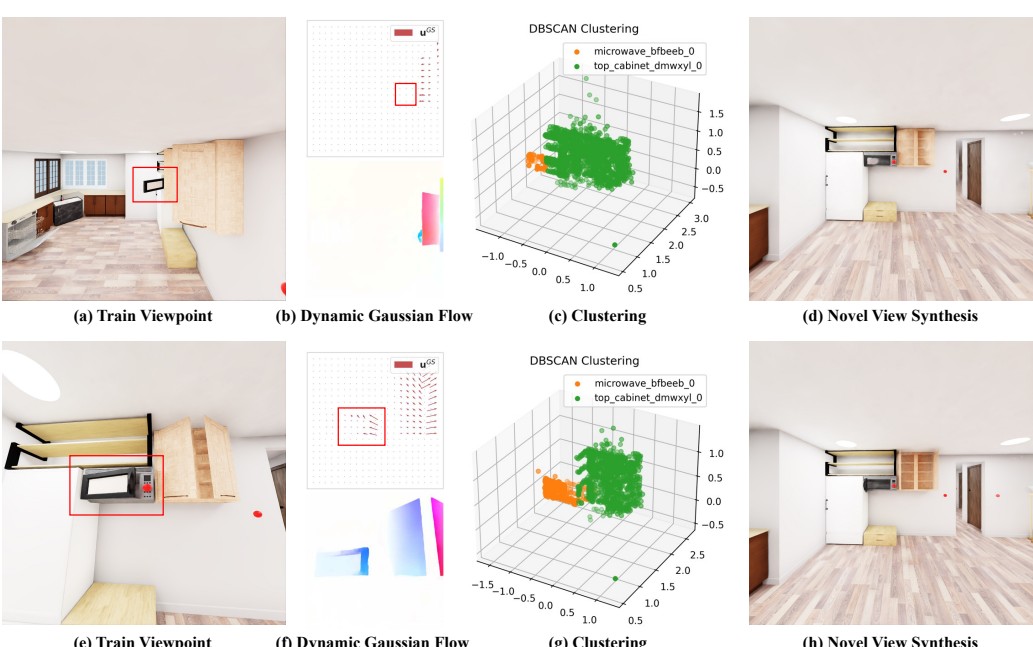

Figure 16: Visualization of dynamic gaussian flow, clustering results, and novel view synthesis under conditions of incomplete moving areas.

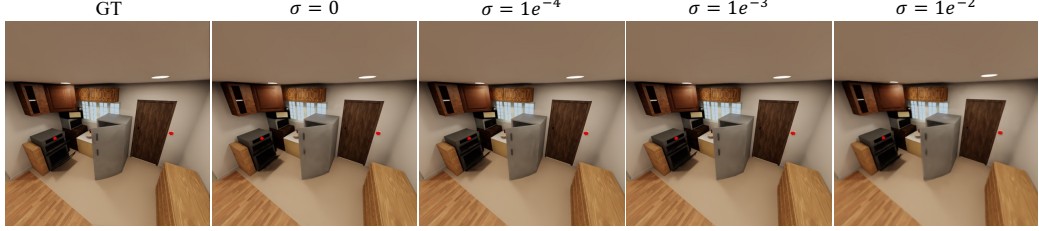

Figure 17: Qualitative results on #seq001 with noisy camera poses.

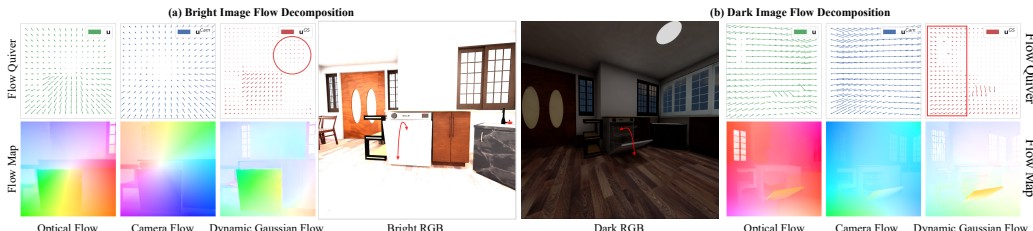

Figure 18: Failure cases due to excessively intense or insufficient lighting.

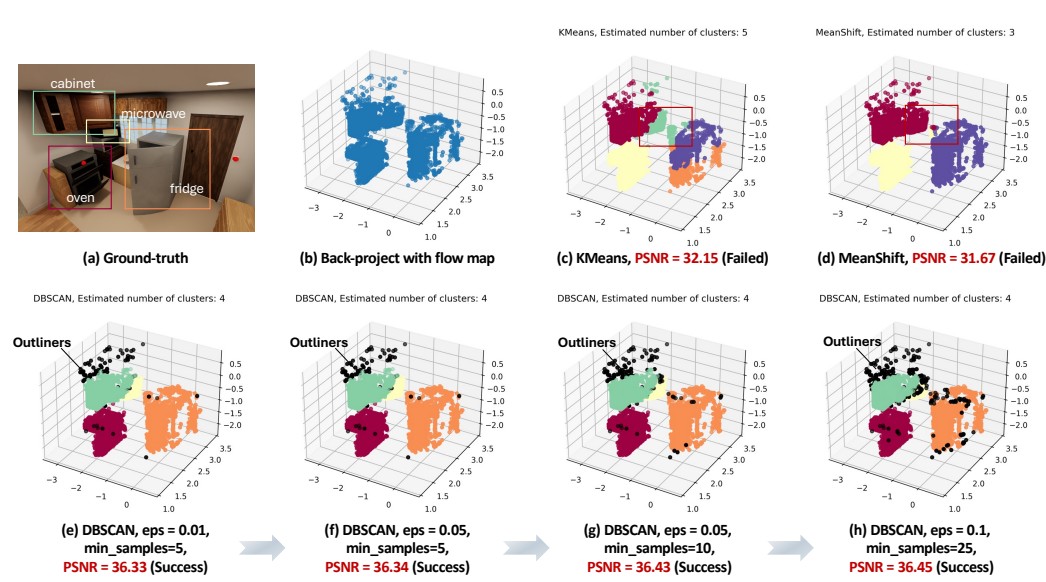

Figure 19: Comparison of clustering results among KMeans, MeanShift and DBSCAN with varying parameters on #seq001 of OmniSim.

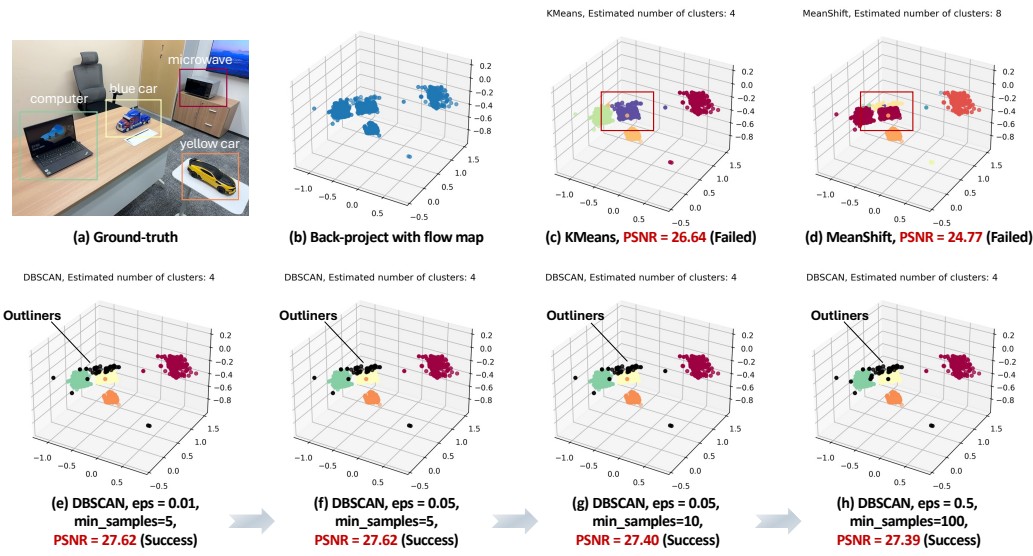

Figure 20: Comparison of clustering results among KMeans, MeanShift and DBSCAN with varying parameters on #seq008 of InterReal.

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
