# OpenReview forum: "FreeGaussian: Guidance-free Controllable 3D Gaussian Splats with Flow Derivatives"
_ICLR.cc/2025/Conference — Submitted to ICLR 2025_

### Official Review · Reviewer_93CX · 2024-11-02

**Soundness:** 2
**Presentation:** 2
**Contribution:** 2
**Rating:** 6
**Confidence:** 4

**Summary:**

This work focuses on reconstructing scenes from monocular videos and enabling controllable editing of Gaussian motion. First, the authors decouple motion information from optical flow to constrain the Gaussian flow. Then, they use DBSCAN to aggregate Gaussian trajectories and incorporate a 3D control vector for trajectory editing.

**Strengths:**

1. The figures in the paper are sufficiently clear. From the qualitative results, the performance surpasses that of the compared methods.
2. The motivation of the paper is clear; reconstructing dynamic scenes from monocular videos and editing motion is meaningful.
3. The authors conducted experiments on both synthetic and real scenes and provided comparisons with recent works.

**Weaknesses:**

1. The expression "self-supervised optimization" in line 18 is inaccurate; in this work, the supervision of Gaussian flow comes from an additional optical flow estimator.
2. The writing of the paper needs improvement. Section 3.2 is a bit hard to follow. Additionally, there are some minor errors, such as "caculated" in L280 and "conducted" in L299. Some papers, such as Gaussianflow and 3DGS, have been repeatedly cited.
3. The relationship between the two main contributions in the paper is unclear. The constraint on Gaussian flow and the control of motion seem to have no direct connection. The authors need to clarify this to highlight the relevance of these two parts.
4. There is a lack of necessary explanation regarding "3D Spherical Vector". Why can controllable editing of trajectories be achieved without any constraints during the control phase? If only the original trajectories are trained during the training phase, why can a "manually input interactive 3D vector" be used in the control phase?
5. The proposed method seems to show only a slight improvement compared to LiveScene, and may even perform worse than LiveScene in some sets.

**Questions:**

1. In Section 3.2, why predict camera flow using "translation velocity v and rotational velocity ω" instead of the relative camera pose R, T? What is the difference between it and the concurrent paper MotionGS [1] ?
2. In L251, how does the predicted Gaussian flow identify dynamic 3D Gaussians through back-projection? Since flow maps are in 2D space and Gaussians exist in 3D space, how is this mapping relationship determined? Is the DBSCAN clustering distance-based?
[1] Zhu R, Liang Y, Chang H, et al. MotionGS: Exploring Explicit Motion Guidance for Deformable 3D Gaussian Splatting. NeurIPS 2024.

---

> ### Author Response · Authors · 2024-11-20
>
> > Q1. Fix expression, typo and citations in the paper.
> >
>
> Thanks. We have carefully revised the manuscript according to your comments. “self-supervised optimization” was originally used to highlight FreeGaussian's annotation-free nature, thus we have updated the terminology to **“annotation-free optimization**” as  suggested. **Sec 3.2** **of the manuscript** analytically decoupled dynamic Gaussian flow from optical flow and camera motion via differential analysis with alpha composition. We include the detailed derivations in the supplementary materials to preserve the brevity and clarity of the manuscript. We kindly invite the reviewer to refer to **Sec 6 of the supplementary** for further information.
>
> > Q2. The relevance between the constraint on Gaussian flow and the control of motion.
> >
>
> **The constraint on Gaussian flow enables the separation of object motion from camera motion by decomposing optical flow into dynamic Gaussian flow and camera flow, which is essential for identifying controllable regions and achieving independent control over object motion in dynamic scenes.**
>
> We reiterate our motivation and the contributions, as well as the relationships between them. We are committed to reconstructing controllable scenes with multiple objects and achieving independent control over different objects in dynamic scenes. For all the controllable scene datasets like [2], there are complex camera motions and multiple object motions. To achieve independent control over objects, we must satisfy two fundamental requirements: firstly, identify all controllable regions in the scene, and secondly, separate the objects in the scene. To identify interactable regions, we utilize an optical flow estimator to detect changing regions in the scene. Given that the optical flow is a composite of object motions and camera motion, we must employ a complex flow decomposition process to separate it into two distinct components: dynamic Gaussian flow and camera flow. Here, the dynamic Gaussian flow is generated by the object motion itself, which is essential for motion control. After determining the controllable regions in the scene, we establish a correspondence between the controllable regions and their corresponding 3D gaussian ellipsoids through back projection. Finally, by modeling the trajectories of the gaussian ellipsoids corresponding to the controllable regions, we achieve control over object motion.

---

> ### Author Response · Authors · 2024-11-20
>
> > Q3. The explanation of "3D Spherical Vector".
> >
>
> The introduction of “3D Spherical Vectors” has significant implications. Previous methods[1][2] perform scene control by mapping the object motion state to a 1D control signal and perform control over the object by varying this 1D state vector. However, modeling such a 1D signal requires strong prior knowledge like the object state at each time step[1][2] or need to approximate the motion trajectory using PCA[9]. Without the control signal supervision during training, these models collapse, failing to reconstruct and losing scene control capabilities. We overcome this limitation by representing the Gaussian states with 3D spherical vectors, which can be directly obtained from dynamic Gaussian tracking trajectory. As shown in **Fig.1 of the manuscript**, during the training stage, we represent the Gaussian dynamics state using cluster trajectory coordinates and jointly train the network to model Gaussian dynamics. In the control stage, we manually input interactive 3D vector*,* retrieving the Gaussian dynamics from the network. This technique eliminates the requirement of control signals and curve fitting while increasing control flexibility.
>
> > Q4. Why can controllable editing of trajectories be achieved without any constraints during the control phase?
> >
>
> In our implementation, control over the object is achieved by querying the dynamic Gaussian trajectories. We store the motion trajectories of each Gaussian ellipsoid in advance and search for the nearest point on the original trajectory given the manually input vector in the control stage. This technique allows FreeGaussian to achieve "unconstrained" control over the object, as the control vector will always match the nearest point on the original trajectory.
>
> > Q5. Why can a "manually input interactive 3D vector" be used in the control phase?  If only the original trajectories are trained during the training phase.
> >
>
> There exists a certain mapping relationship between the 3D spherical vector and the motion trajectory of the object's Gaussian ellipsoids, as formulated in **Eq.(6) of the manuscript**. During the training phase, we track the motion trajectory of the 3D Gaussians and encode the trajectory $\mathbf{v}_c^i$ , which is fed into a lightweight model $\Theta$ to predict Gaussian dynamics $(\Delta\mathbf{X}_i, \Delta\mathbf{\Sigma}_i)$. In the control stage, a manually input 3D spherical vector, serving as a trajectory $\mathbf{v}_c^\prime$, is fed into the network to obtain the dynamics of 3D gaussian ellipsoids. Finally, we query on the original motion trajectory, which is consistent with what is described in Q4.
>
> > Q6. The proposed method seems to show only a slight improvement compared to LiveScene[2].
> >
>
> It is essential to emphasize that **our method is completely annotation-free**, relying solely on an existing optical flow estimator to generate 2D optical flow as pseudo-GT, along with image RGB as real GT. In contrast, **LiveScene depends on masks and dense control variables** during training to perceive and control objects within the scene.
>
> By **eliminating the need for dense annotations and still achieving better performance**, our method demonstrates significant effectiveness and superiority. Furthermore, it offers a viable solution for **unsupervised, large-scale scenario reconstruction and control**, highlighting that FreeGaussian is both innovative and promising. This advancement opens new avenues for research and application where annotated data is scarce or unavailable.

---

> ### Author Response · Authors · 2024-11-20
>
> > Q7 The benefits of using “translation velocity v and rotational velocity ω” and the difference between it and MotionGS[10].
> >
>
> Since MotionGS was not publicly available before the ICLR paper submission deadline of Oct 01 '24, direct comparison is unfair. We had not seen the MotionGS before the review period and only cited GaussianFlow. Nevertheless, we're glad to discuss the difference between MotionGS and our algorithm's in flow derivation and the benefits of ours. We will cite the article accordingly.
>
> Detailed dynamic Gaussian flow analysis can be found at **Sec 6 of the supplementary.** FreeGaussian is formulated under instantaneous motion assumption. The camera and 3D Gaussians hold separate velocities (both rotation and translation) in consecutive frames 0 and t. In contrast to MotionGS, which solely considers rendering-level depth and computes optical flow through back-projection, we model the complete motion process at the Gaussian ellipsoid level and rigorously derive the optical flow relationship among optical flow  $\mathbf{u}$ , camera flow  $\mathbf{u}^\text{Cam}$, and dynamic gaussian flow $\mathbf{u}^\text{GS}$ under instantaneous motion in Eqs.9-16. According to Possion's equation[4], the rotation and translation velocities $\mathbf{\omega}$ and $\mathcal{v}$ can be defined with $\mathbf{R}^{\top}(t)\dot{\mathbf{R}}(t) = [\boldsymbol{\omega}]_{\times}$, $\mathbf{R}^{\top}(t)\dot{\mathbf{T}}(t) = \boldsymbol{v}$, and $\mathbf{R}^{\top}(t)\dot{\mathbf{T}^\text{GS}}(t) = \boldsymbol{v}^\text{GS}$. Specifically, the relative rotation is modeled using the small rotation approximation $\mathbf{R} = \text{Exp}(\mathbf{\omega}) \approx \mathbf{I} + [\mathbf{\omega}] _ {\times}$*,* where *$[\mathbf{\omega}] _ {\times}$*is the skew-symmetric matrix of rotation velocity $\mathbf{\omega}$.  Note that our derivation is more rigorous, and in this task, since the camera pose is available, our algorithm degenerates to computing rotation and translation velocities using the poses of adjacent frames. Please note the residual term $\Delta$ in **Eq.(9)**; we prove that simply using back-projection in the degenerate case will cause errors, with an error magnitude of  $\Delta$, which is a significant difference from MotionGS in the degenerate case.
>
> Based on our formalization, we find that considering the complete Gaussian motion and 2D optical flow changes from 0 to t at a deeper level of the Gaussian ellipsoid hierarchy leads to higher accuracy, as evidenced by the residual term $\Delta$ shown in **Eq.(9) of the manuscript** and ablation in **Fig.6 of the supplementary**. Furthermore, a more important advantage is that our rigorous formula is more general. MotionGS uses discrete camera poses to compute optical flow through backprojection, whereas we model camera motion using differential rotation and translation velocities, allowing us to compute optical flow at any time between 0 and t using **Eq.(9)**, whereas MotionGS is limited to 0 or t. This difference is similar to that between discrete and continuous functions and has important applications in various computer vision tasks, which model the accurate time interval between frames, such as video frame interpolation, rolling shutter correction, and SLAM[5][6][7]. MotionGS is a degenerate solution of the method we proposed.
>
> We believe the score of this paper should not be influenced by MotionGS, especially when derivation has significant differences in terms of formalization, generality, and algorithm motivation.

---

> ### Author Response · Authors · 2024-11-20
>
> > Q8. How does the predicted Gaussian flow identify dynamic 3D Gaussians through back-projection?
> >
>
> The back projection is implemented by Gaussian rasterization, as detailed in [3]. This process projects 3D Gaussians on the screen and converting a 3D Gaussian into 2D representations with perspective projection. By leveraging the GSplats[8], we acquire the 2D projection of 3DGS. Through the mask of dynamic gaussian flow within the flow map, we establish a correspondence between 2D pixel and 3D Gaussians, subsequently linking the dynamic flow to specific spatial elements in the 3D space.
>
> > Q9. Is the DBSCAN clustering distance-based?
> >
>
> Yes, we employ the Euclidean distance between the center of 3D Gaussians as the distance metric in the DBSCAN algorithm. as mentioned in **Line 304**. More detailed implementation can be found in **Sec 4 Implementation details of the manuscript**.
>
> We appreciate the reviewer's suggestions. We hope that our revisions, details explanations, and comparisons with MotionGS can increase the likelihood of a higher score.
>
> **Reference**
>
> [1] CoNeRF: Controllable Neural Radiance Fields
>
> [2] LiveScene: Language Embedding Interactive Radiance Fields for Physical Scene Rendering and Control
>
> [3] 3D Gaussian Splatting for Real-Time Radiance Field Rendering
>
> [4] Subspace methods for recovering rigid motion I: Algorithm and implementation
>
> [5] Fast Rolling Shutter Correction in the Wild
>
> [6] Linear differential algorithm for motion recovery: A geometric approach
>
> [7] FILM: Frame Interpolation for Large Motion
>
> [8] gsplat: An Open-Source Library for Gaussian Splatting
>
> [9] CoGS: Controllable Gaussian Splatting
>
> [10] MotionGS: Exploring Explicit Motion Guidance for Deformable 3D Gaussian Splatting

---

> ### Author Response · Authors · 2024-11-29
> **Follow up process**
>
> Thanks for your time and effort in reviewing our work. We want to kindly follow up on the review process and greatly appreciate your feedback. Look forward to your response.

---

### Official Review · Reviewer_rRqp · 2024-11-03

**Soundness:** 3
**Presentation:** 3
**Contribution:** 3
**Rating:** 5
**Confidence:** 4

**Summary:**

The paper addresses the reliance on manual annotation and the need for masking in the Gaussian Splatting scene control line of work. The paper presents a simple intuition that the controllable dynamic regions can be identified from optical flow maps capturing both camera motion and scene motion. The dynamic regions are then clustered using DBSCAN to alleviate computation and ensure uniformity in motions. 3D spherical vectors are used for controls in replacement of commonly employed 1D state variable.

**Strengths:**

The paper's motivation is clearly stated. The paper presents a simple yet effective intuition of isolating the controllable region via camera motion disentanglement from optical flow maps. The motivation for adopting a 3D spherical vector for the control signal is discussed in detail. The diagrams of the overall system are clear and easy to understand, and the experiments sufficiently support the authors' arguments.

**Weaknesses:**

The paper is overall nicely written and properly presented. One main concern is the robustness of the proposed control region isolation and flow guidance in monocular captures. Many monocular captures do not have reliable camera poses, which can cause an issue with the flow-based isolation strategy presented in the paper. DBSCAN could also produce undesired clustering results in scenes if the Gaussian distributions are uneven across the scene, eg. when some regions are denser or sparser. It would be reasonable to show some visual comparisons and quantitative results on HyperNeRF scenes and CoNeRF face scenes as well.

**Questions:**

The paper can be further improved with the following clarifications:

1. As discussed in the Weakness section, qualitative and quantitative results on datasets with noisy camera poses would help illustrate the robustness of the proposed method.
2. The paper currently does not show any failure cases. When would the approach fail? The limitation section mentions that lighting variations would compromise the results. However, this is not sufficiently supported in the discussions. How much of a lighting variation would affect the method and to what extent?

---

> ### Author Response · Authors · 2024-11-20
>
> > Q1. Qualitative and quantitative results on datasets with noisy camera poses.
> >
>
> When the camera pose is relatively noisy, the model's performance is indeed affected. In our implementation, we employed the pose optimizer during the training, which mitigates the issues caused by noisy poses. Specifically, the poses are optimized by the Adam, with an initial learning rate of 1e-4 and an exponential decay scheduler.
>
> In our experiments, we evaluate on two real-world datasets, namely CoNeRF Controllable and InterReal. The poses of these datasets were obtained through COLMAP and inherently possess certain levels of noise. **Tab.1 and Tab.3 of the manuscript** demonstrate the superiority of our method over other methods in terms of rendering metrics such as PSNR, highlighting the robustness of our method in the presence of noisy poses. **Fig.7 and Fig.13 of the supplementary** show the visualization comparisons for these two datasets, from which it can be observed that our method obtain more detailed and accurate representations of the objects. In contrast, other methods failed to capture the object's shape and resulted in significant artifacts.
>
> To illustrate the robustness of the proposed method, we also conduct a new experiment on #seq001 of OmniSim with manual noise, which is generated using a Gaussian distribution with a mean of 0 and a variance of σ. The table below shows the impact of σ ranging from 1e-4 to 1e-2 on the rendering metrics. It shows that FreeGaussian still demonstrates strong rendering and control capabilities at lower noise levels. As the noise increases, the model's performance gradually deteriorates, but still maintains at a certain level. **Fig.17** **of the supplementary** also provides a good illustration of this process. When σ is 1e-2, the controllable parts like the door become blurry.
>
> | variance σ | 0 | 1e-4 | 1e-3 | 1e-2 |
> | --- | --- | --- | --- | --- |
> | PSNR | 36.34 | 36.04 | 35.41 | 28.31 |
> | SSIM | 0.980 | 0.980 | 0.976 | 0.91 |
> | LPIPS | 0.046 | 0.046 | 0.049 | 0.152 |
>
> > Q2. Could undesired clustering produced by DBSCAN affect the following process?
> >
>
> Yes. In the case of extremely close objects, the clustering effect may not be ideal. Please refer to **Fig.5 of the manuscript**, the microwave and top cabinet in the scene are very close to each other, resulting in unclear clustering boundaries between these two objects. Nevertheless, clustering centers still fall within the dynamic range of the corresponding objects, and FreeGaussian achieves satisfying rendering results in the control stage.  The robustness of FreeGaussian comes from its polling operation, which weights all Gaussian trajectories within a control region, rather than treating each as an independent control variable. The rendering metrics in **Tab.5 of the manuscript** support its effectiveness.

---

> ### Author Response · Authors · 2024-11-20
>
> > Q3. Failure cases illustration.
> >
>
> As previously discussed in Q1, noise introduced by camera pose affects the model's performance, and we indeed showcased corresponding failure cases, as seen in row 5 in **Fig.7 of the supplementary**. The figure shows that although the mechanical dog has a larger residual shadow, our reconstruction result is still relatively better compared to other methods, despite being a failure case. In addition to camera pose, rapid object motion is another factor causing the model to fail in reconstructing the scene. To illustrate this, we create an interactable scene in the OmniGibson simulator[1] and manipulate a chest with extremely fast speed. **Fig.14(b)** shows the rendering results of our model, which shows that rapid motion leads to a failure of our first-stage model, yielding a blurry boundary. Despite successfully decoupling moving objects, our initial 4D pre-trained model failed to capture the motion trajectories of 3D gaussian ellipsoids, thereby hindering the completion of subsequent control tasks.
>
> > Q4. How much of a lighting variation would affect the method and to what extent.
> >
>
> Thanks for this question. Since our method relies on off-the-shelf flow estimator, the accuracy of optical flow estimation will significantly impact our model's performance. Existing flow estimators are susceptible to environmental and lighting conditions, and therefore, we investigate the effects of overly strong and weak lighting on optical flow and dynamic Gaussian flow. **Fig.18 of the supplementary** shows that due to uneven lighting, the flow estimator overestimates the environmental flow, resulting in corresponding high-brightness regions in non-moving areas. The inaccuracy of flow estimation affects the clustering results and ultimately influences the final control process.
>
> We appreciate the reviewer's attention to the robustness of our model and hope that the additional experiments we have provided can demonstrate the robustness and superiority of our model, and we look forward to a higher score.
>
> **Reference:**
>
> [1] BEHAVIOR-1K: A Benchmark for Embodied {AI} with 1,000 Everyday Activities and Realistic Simulation

---

> ### Comment · Reviewer_rRqp · 2024-11-25
>
> Thank you for the explanations. Following up on the issue with DBSCAN instabilities since using DBSCAN for initializing the controls is one of the main contributions yet the current explanation does not thoroughly cover the pros and cons of this design choice. As many prior works on point cloud clustering have pointed out, the hyperparameter tuning for the DBSCAN is a major issue since it depends on factors like scene scale, local densities, and local manifold geometry. How is the hyperparameter tuning being handled in your approach? Were the hyperparameters manually tuned for the experiments or was there any tuning method employed and tailored for Gaussians? Besides DBSCAN there are also many other clustering methods, as explored in other GS control/editing frameworks like Di-MiSo - can you explain why DBSCAN is selected over alternatives and how it might be more suitable over alternatives for real/synthetic scenes?

---

> > ### Author Response · Authors · 2024-11-26
> >
> > We appreciate the reviewer’s insightful question regarding DBSCAN. Additional illustrations are provided in [Fig.19 Line 1371 and Fig.20 Line 1398](https://openreview.net/pdf?id=rQV33MVNWs#page=10.19).
> >
> > Clarify misunderstandings:
> >
> > - Firstly, **DBSCAN is not the contribution of FreeGaussian**. Any clustering technique could be seamlessly integrated into our methods. Our primary contribution lies in exploring interactable structures with flow priors and restoring scene interactivity without any manual annotations.
> > - Secondly, the clustering is robust. Based on the clean and low-overlapped back-projected Gaussians, DBSCAN just simply classifies the sparse distributed Gaussians, as illustrated in Fig.19(b) and Fig.20(b).
> > - Is Di-MiSo publicly available? It cannot be found online. **The most related work is D-MiSo in paper**[1]. However, we do not find any clustering techniques in D-MiSo. The key idea of D-MiSo is to use multi-Gaussian components, consisting of core Gaussian and sub-Gaussian, for modeling dynamic scenes. It does not use any clustering algorithm to identify dynamic regions in the scene.
> > - Furthermore, D-MiSo is primarily evaluated on datasets with a single primary object, such as D-NeRF and DyNeRF. In contrast, our method tackles more challenging scene-level tasks, identifying and clustering dynamic regions into specific interactable objects.
> >
> > Why DBSCAN?
> >
> > Compared with widely used clustering methods, such as KMeans, **DBSCAN is more robust to noise with outliner handling and more flexible without predefined cluster numbers**. Besides, MeanShift clustering may converge to local optima depending on the cluster landscape and initial window locations. We conduct additional experiments to illustrate the effectiveness of DBSCAN, and the impact of scene scale, local densities, and local manifold geometry.
> >
> > The first experiment is conducted on #seq001 of the OmniSim dataset, with a cabinet and microwave close to each other, in a small-scale kitchen room. **Fig.19 Line 1371** shows the clustering results of DBSCAN under different hyperparameters, as well as comparisons with KMeans and MeanShift. The results in **Fig.19 (e)-(h)** demonstrate that, DBSCAN exhibits remarkable stability and accuracy, even when the hyperparameters fluctuate from (eps = 0.01, min_samples=5) to (eps = 0.1, min_samples=25). DBSCAN effectively captures the geometry of objects and obtains accurate outliers, with clear contours for the dishwasher and fridge. In contrast,  KMeans causes a large number of noisy points as part of the objects (**Fig.19 (c)**) and Meanshift fails to guarantee an appropriate number of clusters (**Fig.19 (d)**), leading to unwanted changes in the environment and negatively impacting the rendered results.
> >
> > In **Fig.20 Line 1398**, we conduct another experiment on the #seq008 of the InterReal dataset, a larger room with a more expansive scene scale. Despite the larger scale, DBSCAN maintains robustness, accurately segmenting out all 4 objects with the default hyperparameters. While KMeans identifies four clusters, it contains misclassified points and significant noise. MeanShift, on the other hand, produces incorrect 8 cluster centers.
> >
> > Hyperparameters tuning.
> >
> > Hyperparameters are manually set and the default parameters (eps = 0.05, min_samples=5) are applicable to most datasets in this paper. Besides, a fixed ratio (typically 10%) of the total points is set to limit the cluster size, which ensures the clusters are of manageable and adaptive size.
> >
> > In summary, **DBSCAN is robust to scene scale, effective in high-density regions, and retain the inherent geometrics** of the objects, outperforming other widely used algorithms.  With slight tweaks to the eps and min_samples, the DBSCAN algorithm works well.
> >
> > **Reference:**
> >
> > [1] D-MiSo: Editing Dynamic 3D Scenes using Multi-Gaussians Soup

---

> ### Author Response · Authors · 2024-11-29
> **Follow up process**
>
> Thanks for your time and effort in reviewing our work. We want to kindly follow up on the review process and greatly appreciate your feedback. Look forward to your response.

---

### Official Review · Reviewer_j2YT · 2024-11-03

**Soundness:** 3
**Presentation:** 3
**Contribution:** 3
**Rating:** 6
**Confidence:** 4

**Summary:**

This work leverages dense optical flow and changes in camera motion trends to perceive dynamic regions. By decoupling the linear and angular velocities of the moving camera and the linear velocity of dynamic scene Gaussian primitives, it achieves dynamic scene reconstruction from a moving perspective. The method realizes unsupervised, semantic- and text-prior-free 4D reconstruction, offering a novel approach for unsupervised Gaussian physical simulation.

**Strengths:**

1. The integration of optical flow and camera motion for perceiving dynamic regions is compelling, and the mathematical decoupling of dynamic Gaussian motion and camera movement is innovative.

2. Despite some jagged artifacts in local areas, the overall FreeGaussian reconstruction shows impressive 4D reconstruction quality, enhancing 3DGS robustness for dynamic scenes. Dynamic segments can simulate real motion trajectories for dynamic simulation through Gaussian modeling.

3. The work proposes a differentiable rendering constraint for optical flow and camera motion, decouples the camera’s rotation, translation, and dynamic Gaussian velocity, and solves all motion states using gradient descent. This is integrated into Gaussian’s CUDA implementation, enabling dynamic object rendering from moving perspectives based on Gaussian primitives.

4. The paper is well-written, with comprehensive experiments, and the mathematical derivation in the rendering section is rigorous.

**Weaknesses:**

1. Could there be instances where the perceived region is incomplete? For instance, variations in optical flow and camera motion trends can detect dynamic areas, but due to frequent object occlusions and limited observational perspectives, it might not be possible to view the entirety of dynamic regions from multiple perspectives. Could this impact the method’s effectiveness?

2. DBSCAN requires additional parameter design to cluster Gaussians. When objects are closely spaced, the spatial distribution of Gaussian primitives might lead to inaccurate clustering results.

3. The optimization for pre-trained Gaussians in the first stage lacks sufficient geometric priors, potentially leading to inadequate high-quality scene rendering in some scenarios, which may impact subsequent reconstruction quality.

**Questions:**

1. The design of Eq. (6) lacks details on the specific activation functions and output processing used.

2. Could the pre-trained Gaussians in the first stage have suboptimal dynamic reconstruction, potentially impacting the final simulation results?

3. The reconstruction of Gaussians does not incorporate advanced geometric constraints, which might affect DBSCAN accuracy in later stages.

4. The loss design is under the assumption of Gaussian isotropy. However, pre-trained Gaussians seem not to include isotropy constraints, which might significantly impact this proposition and its efficacy. I find this theory quite intriguing.

5. Following the previous question, if it does indeed cause an impact, then perspective transformations A and B will be introduced in L_uGS, and the CUDA part of Gaussian cannot directly backpropagate the gradients to the linear transformations A and B of the motion state variables. Is it necessary to additionally write the Jacobian matrix for backpropagation in CUDA?

---

> ### Author Response · Authors · 2024-11-20
>
> > Q1. Could there be instances where the perceived region is incomplete? Could this impact the method’s effectiveness?
> >
>
> In all our evaluation sequences, although certain objects may be occluded from specific viewpoints, the moving regions are always complete when considering the entire video sequence. To investigate our model's performance under incomplete moving region conditions, we conduct a new experiment.
>
> As illustrated in **Fig.16(a) of the supplementary**, from the illustrated camera view, the microwave is occluded by the cabinet. In this case, the perception of the microwave through the decomposed dynamic Gaussian flow is relatively weak, leading to poor clustering results, as shown in **Fig.16(b)**, where only a part of the microwave is obtained. When we move the camera to a viewpoint where the microwave is visible, the rendering result is poor, with only partial regions undergoing motion, see **Fig.16(d)**.
>
> However, when the training sequences include both incomplete (**Fig.16(a)**) and complete (**Fig.16(e)**) viewpoints, a good clustering result for the microwave could be obtained, as shown in **Fig.16(g)**, where the overall geometric structure of the microwave is relatively clear. The rendering result is satisfactory in novel view synthesis (**Fig.16(h)**).
>
> This demonstrates that FreeGaussian can correctly control and render in novel views even when the training includes incomplete moving regions, showcasing its robustness and superiority over current Nerf-based and Splat-based methods.
>
> > Q2. Could DBSCAN result in inaccurate clustering when objects are closely spaced?
> >
>
> Yes. In the case of extremely close objects, the clustering effect may not be ideal. Please refer to **Fig.5 of the manuscript**, the microwave and top cabinet in the scene are very close to each other, resulting in unclear clustering boundaries between these two objects. Nevertheless, clustering centers still fall within the dynamic range of the corresponding objects, and FreeGaussian achieves satisfying rendering results in the control stage. The robustness of FreeGaussian comes from its polling operation, which weights all Gaussian trajectories within a control region, rather than treating them as an independent control variable. The rendering metrics in **Tab.5 of the manuscript** support its effectiveness.
>
> We conduct an additional experiment to illustrate the the parameter setting and the effectiveness of DBSCAN, as shown in **Fig.19 of the supplementary**. The experiment is conducted on #seq001 of the OmniSim dataset, with a cabinet and microwave close to each other in a small-scale kitchen. **Fig.19 Line 1371** shows the clustering results of DBSCAN under different hyperparameters, as well as comparisons with KMeans and MeanShift. In the experiment, a fixed ratio (typically 10%) of the total points is set to **limit the cluster size**, which ensures the clusters are of manageable and adaptive size. The results in Fig.19 (e)-(h) demonstrate that, DBSCAN exhibits remarkable stability and accuracy, even when the hyperparameters fluctuate from **(eps = 0.01, min_samples=5)** to **(eps = 0.1, min_samples=25)**. DBSCAN effectively captures the geometry of objects and obtains accurate outliers, with clear contours for the dishwasher and fridge.
>
> > Q3. The optimization for pre-trained Gaussians in the first stage lacks sufficient geometric priors, potential impact DBSCAN accuracy and the final result.
> >
>
> Notably, Gaussian Pre-training aims to perceive the dynamic regions in the scene, i.e., the controllable objects in the scene. **For dynamic scene reconstruction, 4DGS methods still achieve high-quality reconstruction without prior knowledge**, such as depth or point cloud with geometric shapes, making it satisfying for FreeGaussian’s first-stage training. For more complex motions, we introduce the Dynamic Gaussian Flow Loss in **Sec 3.4 of the manuscript**, which utilizes optical flow and camera flow as priors to achieve more robust and accurate modeling capabilities. Dynamic Gaussian Flow Loss helps to smooth the underlying 3DGS optimization process and further enhances the 4D scene reconstruction quality of 4DGS. The ablation in **Sec 4.4** supports this.
>
> Besides, in the second stage, **FreeGaussian weights all Gaussian trajectories within a control region**, rather than treating them as an independent control variable, which improves robustness, and mitigates DBSCAN clustering inaccuracies. Finally, we must emphasize that joint training in the second phase optimizes not only 3D spherical vectors but also 3D Gaussians and Hash Features, alleviating the problem of training in the first stage.

---

> ### Author Response · Authors · 2024-11-20
>
> > Q4. The detailed design on the specific activation functions and output processing used of Eq.(6).
> >
>
> We have modified **Eq.(6) of the manuscript** to $\boldsymbol{f}_{\Theta}(\mathbf{E}(\mathbf{v}_c^j,  \mathbf{X}_i)) \mapsto \left \langle \Delta\mathbf{X}_i, \Delta\mathbf{\Sigma}_i \right \rangle$, which describes the mapping relationship between the 3D spherical vector and the motion trajectory of the object's Gaussian ellipsoids. During the training phase, we track the dynamic Gaussian trajectory, and encode the trajectory $\mathbf{v}_c^i$ , and Gaussian coordinates using Hash encoding. The encoded features are then fed into a lightweight MLP $\Theta$, which predicts the offsets of the Gaussians. The hidden layers of the MLP utilize ReLU as an activation function. In the control phase, a manual 3D spherical vector as well as the 3D Gaussians are input to the network and the dynamics of the 3D Gaussian ellipsoids are predicted. Finally, based on the prediction, we query the original motion trajectory as the final output.
>
> > Q5. Does the inconsistency in the assumption of Gaussian isotropy between the two stages impact the subsequent performance?
> >
>
> By following existing works[1][2][3], we assume isotropy between two consecutive frames, solely to simplify the loss term, enhance numerical stability, and accelerate back-propagation. However, under instantaneous motion, the Gaussian ellipsoid changes minimally, resulting in a similarly small difference as $\Delta\mathbf{u}^\text{GS} = \| \mathbf{u}^\text{GS} - \mathbf{\tilde{u}}_i^\text{GS} \|$. Although providing a clear analysis from a numerical perspective is challenging, we have implemented the complete loss function in CUDA and set up several experiments to compare the actual effects. The experimental results show that using the complete loss function significantly increases the computation time, but does not yield a notable improvement in rendering quality. Therefore, it is reasonable to consider ignoring $\Delta\mathbf{u}^\text{GS} = \| \mathbf{u}^\text{GS} - \mathbf{\tilde{u}}_i^\text{GS} \|$. Furthermore, we agree that introducing the concept of isotropy is only partially accurate, and we plan to replace this isotropy assumption with a detailed experimental analysis.
>
> |  | PSNR | SSIM | LPIPS | FPS |
> | --- | --- | --- | --- | --- |
> | Ours | 35.31 | 0.975 | 0.062 | 123.88 |
> | Ours w/o isotropy assumption | 35.26 | 0.972 | 0.067 | 50.22 |
>
> > Q6.  Gradients backpropagation to the linear transformations A and B. More details about the CUDA implementation.
> >
>
> Please recall the pipeline in **Line 115** and CUDA implementation in **Line 787**, we aim to optimize the Gaussian and neural network through back-propagation using $\mathcal{L}_\text{uGS}$. Note that in **Eq.(3)** and **Eq.(4) of the manuscript**, although the linear transformations A and B are already present in the loss function, they do not directly participate in the optimization. Instead, we decouple the Gaussian flow via **Eq.(3)**, which serves as the ground truth in the loss function calculation. Furthermore, the challenge in CUDA optimization does not lie in computing the Jacobians of A and B. Rather, it is to establish a mapping relationship between image pixels and 3D Gaussians, enabling the retrieval of all relevant Gaussian ellipsoids through pixel coordinates, thereby ensuring that the optical flow loss can be computed and propagated.
>
> We sincerely thank the reviewer, one of the few who have carefully read and thoughtfully commented on our paper. We look forward to an improved and objective score, free from misunderstandings.
>
> **Reference**
>
> [1] SplaTAM: Splat Track & Map 3D Gaussians for Dense RGB-D SLAM
>
> [2] Align Your Gaussians: Text-to-4D with Dynamic 3D Gaussians and Composed Diffusion Models
>
> [3] GaussianFlow: Splatting Gaussian Dynamics for 4D Content Creation

---

> ### Author Response · Authors · 2024-11-29
> **Follow up process**
>
> Thanks for your time and effort in reviewing our work. We want to kindly follow up on the review process and greatly appreciate your feedback. Look forward to your response.

---

### Official Review · Reviewer_jysV · 2024-11-03

**Soundness:** 2
**Presentation:** 2
**Contribution:** 2
**Rating:** 3
**Confidence:** 3

**Summary:**

The paper introduces a method to reconstruct controllable Gaussian splats from a monocular video. The contributions include a derivation of 2D scene flow (referred to as dynamic gaussian flow) rendering and a interactive control pipeline that relies on spatial clustering. The method is tested on a few synthetic datasets (CoNeRF, OmniSim, InterReal).

**Strengths:**

- The task of improving the controllability of gaussian splatting is interesting.
- The idea of using spatial clustering to obtain controllable objects is sound.
- The derivation of 2D scene flow from 3D scene flow and camera motion looks correct.
- The paper did extensive experiments and comparisons.

**Weaknesses:**

Although some components of the paper is interesting, I have major concerns about the writing and soundness.
- Many technical pieces of the method is missing, which makes it difficult to understand the design choices.
    - What is the purpose of factorizing 2D optical flow to 2D scene flow and camera flow? This make both the derivation and flow loss more complicated than Gao et al 2024 and I don't see the benefit.
    - More algorithmic details about Sec 3.3. would help. What is the distance metric used in DBSCAN algorithm. How is 2D scene flow useful there? What is the model $\Theta$ and what is ${\bf E}$?

- For the flow derivation, I don't see much technical novelty compared to Gao et al 2024. This paper only considers translation of 3D gaussians, but not rotation.

Experiments
- The paper is only evaluated on synthetic data. How does it work (even qualitatively) on real datasets, such as the ones used in Gao et al 2024?

**Questions:**

- Does the datasets used in the paper contain videos of dynamic objects and scenes? If so, how does baselines that only handle static scenes, such as NeRF, 3DGS work?
- If the scene is static, would dynamic gaussian flows become always zeros.

---

> ### Author Response · Authors · 2024-11-20
>
> > Q1. The purpose of factorizing 2D optical flow and it’s benefit compared to Gao et al.[1].
> >
>
> We must emphasize that our task is to reconstruct a multi-object controllable scene and control the objects within the scene. All evaluation sequences contain both camera motion and object motion simultaneously, thus the 2D optical flow consists of two components: one caused by camera motion and the other by object motion itself, as illustrated in **Fig.3 of the manuscript**. Decomposing the optical flow allows us to obtain the changes solely caused by object motion, which serves as the input for subsequent control. In contrast, setting of Gao et al., is relatively simple, with a fixed camera and only a single moving object in the scene, which is suitable for traditional 4D reconstruction. It would fail to control the scene when camera motion is involved, as it mistakenly treats camera motion as object motion. Therefore, the Gaussian flow in Gao et al 2024 is actually a special case of ours. By correctly decomposing camera and object motion, FreeGaussian is able to reconstruct controllable scene and achieve independent object control. Please refer to Q4 for more novelty comparisons.
>
> > Q2. Details about DBSCAN algorithm and the usefulness of dynamic Gaussian flow (2D scene flow).
> >
>
> We employ the Euclidean distance between center of 3D Gaussians as the distance metric in DBSCAN algorithm. as mentioned in **Line 304**. More detailed implementation could be found in **Sec 4 Implementation details of the manuscript**.
>
> As previously explained, dynamic Gaussian flow solely captures the object motions. We leverage this flow to identify the 3D Gaussians associated with the objects. Specifically, after decoupling the dynamic Gaussian flow, a flow map is constructed from it, see **Fig.3(b) of the manuscript** for details. Subsequently, through Gaussian rasterization, we establish a correspondence between the 3D Gaussians and pixels in the image. The flow map then serves as a mask, enabling us to determine the 3D Gaussians matched to the current dynamic Gaussian flow via a splatting process. Once we have identified which Gaussians in 3D space related to the current motion, we cluster the relevant Gaussian ellipsoids, thereby obtaining different cluster centers, i.e., different objects.
>
> > Q3.  The definition of model $\Theta$ and $E$.
> >
>
> In **Eq.(6) of the manuscript**, $E$ is an encoder which encodes the cluster trajectory $\mathbf{v}_c^i$  as well as the coordinates of 3D Gaussians with learnable Hash Grids. The model $\Theta$ is an additional network that maps the 3D Gaussians and the trajectory to variation of control vector, which is used in later control process.

---

> ### Author Response · Authors · 2024-11-20
>
> > Q4. The technical novelty of flow derivation and assertion of neglecting rotation of 3D Gaussians.
> >
>
> Our dynamic Gaussian flow analysis introduces a novel formulation that models both camera motion and dynamic scene motion at the Gaussian ellipsoid level. Unlike Gao et al., who assume a static camera and focus solely on Gaussian motion, our approach derives the optical flow relationship among the observed optical flow $\mathbf{u}$, camera-induced flow $\mathbf{u}^\text{Cam}$, and dynamic Gaussian flow $\mathbf{u}^\text{GS}$. **This comprehensive modeling allows us to decouple the optical flow into components caused by camera motion and by the object’s intrinsic motion, leading to a more accurate and generalizable solution.** Our rigorous derivation under instantaneous motion assumptions is detailed in **Eq.(9) to Eq.(16) of the manuscript** and further elaborated in **Sec 6 of the supplementary**.
>
> While Gao et al. primarily address translation, our method incorporates Gaussian rotation through **Eq.(10)**. Since the self-rotation of isotropic Gaussians does not affect the projection of their centers onto the 2D plane, it is neglected in our formulation. The rotation and translation velocities are defined using Poisson’s equation, as shown in our derivations: $\mathbf{R}^{\top}(t)\dot{\mathbf{R}}(t) = [\boldsymbol{\omega}]_{\times}$, $\mathbf{R}^{\top}(t)\dot{\mathbf{T}}(t) = \boldsymbol{v}$, and $\mathbf{R}^{\top}(t)\dot{\mathbf{T}^\text{GS}}(t) = \boldsymbol{v}^\text{GS}$.
>
> We must emphasize that, the main idea of Gao et al.[1] differs significantly from our proposed dynamic Gaussian flow analysis in terms of **motion formulation, generality, and motivation**. Below are additional explanations in task and method that contribute to the novelty of our approach.
>
> - Gao et al. consider a static camera and do not account for camera motion in their flow derivation. In contrast, our method explicitly models both camera rotation and translation, as well as the rotation and translation of 3D Gaussians. This allows our approach to handle scenarios with significant camera movement, making it more versatile in practical applications.
> - Our approach remains effective in dynamic scenes with camera motion, whereas Gao et al.’s method is limited to static camera setups. In scenarios without camera motion, their method can be viewed as a special case of our more general framework.
> - By modeling the complete motion process at the Gaussian ellipsoid level and considering both camera and object motions, our method achieves higher accuracy in optical flow estimation. The inclusion of the residual term $\Delta$ in **Eq.(9)** and our ablation studies (**Fig.6**) demonstrate the improved performance over methods that do not consider these factors.
> - The proposed loss function in **Eq.(7)** is just one component of our innovative contributions. As discussed in **Lines 235** and **Line 242** of the manuscript, our method leverages dynamic Gaussian flow to explore dynamic Gaussians of interactive objects and extract their trajectories for joint optimization. This enables the restoration of scene interactivity and provides continuous motion constraints for dynamic Gaussian optimization.
>
> We hope this detailed explanation clarifies the technical novelties and key contributions of our work compared to Gao et al. Our method’s ability to model both camera and object motions, consider Gaussian rotations, and automatically discover dynamic objects represents significant advancements in the field. We kindly encourage you to review the relevant sections of our manuscript and supplementary materials for a comprehensive understanding. Thank you for your consideration.

---

> ### Author Response · Authors · 2024-11-20
>
> > Q5. The evaluation dataset and the qualitative results on real dataset.
> >
>
> In our experiments, we employed three datasets, namely CoNeRF consisting of Synthetic and Controllable subsets, OmniSim and InterReal, where **CoNeRF Controllable and InterReal are real-world datasets**, and CoNeRF Synthetic and OmniSim are synthetic datasets. Please refer to [2] and [3] for detailed information about the datasets.
>
> For CoNeRF Controllable and InterReal datasets, **Tab.1 and Tab.3 of the manuscript** provide the corresponding quantitative metrics. Additionally, **Fig.7 of the supplementary** illustrates the visualization results on the InterReal dataset, demonstrating that our model can still reconstruct complex controllable scenes in real-world scenarios.
>
> The following table reports the PSNR metric between our method and others on the DyNeRF dataset. Our results are comparable to Gao's, but significantly outperform other methods.
>
> | Method | Coffee Martini | Cook Spinach | Cut Beef | Flame Salmon | Flame Steak | Sear Steak | Mean |
> | --- | --- | --- | --- | --- | --- | --- | --- |
> | HexPlane | - | 32.04 | 32.55 | 29.47 | 32.08 | 32.39 | 31.70 |
> | K-Planes | 29.99 | 32.60 | 31.82 | 30.44 | 32.38 | 32.52 | 31.63 |
> | MixVoxels | 29.36 | 31.61 | 31.30 | 29.92 | 31.21 | 31.43 | 30.80 |
> | NeRFPlayer | 31.53 | 30.56 | 29.35 | 31.65 | 31.93 | 29.12 | 30.69 |
> | HyperReel | 28.37 | 32.30 | 32.92 | 28.26 | 32.20 | 32.57 | 31.10 |
> | 4DGS | 27.34 | 32.46 | 32.90 | 29.20 | 32.51 | 32.49 | 31.15 |
> | RT-4DGS | 28.33 | 32.93 | 33.85 | 29.38 | 34.03 | 33.51 | 32.01 |
> | Gaussian Flow | 28.42 | **33.68** | 34.12 | 29.36 | **34.22** | **34.00** | **32.30** |
> | Ours | **28.53** | 33.36 | **34.33** | **29.58** | 34.00 | 33.83 | 32.27 |
>
> **Fig.13 and Fig.15 of the supplementary** show more qualitative results on CoNeRF Controllable and DyNeRF datasets. As shown in **Fig.13**, compared to other methods, ours has reconstructed more details than the current SOTA methods. For DyNeRF dataset, as shown in **Fig.15**, our method also has good control over the motion regions. Results demonstrate that our method has certain advantages in real-world scenarios.
>
> Moreover, we have reproduced Gao’s method since the code is not currently available. We compared it with ours on the #seq001, #seq004, and #seq015 subsets of Omnisim, as shown in the **Tab.8 of the supplementary**. From the results, it is evident that in scenarios containing both camera motion and object motion, the advantages of our method become apparent. By decomposing the optical flow, **FreeGaussian isolates the flow generated solely by the object’s intrinsic motion**, enhancing the rendering effect while enabling precise control over the object. In contrast, **Gao’s method treats camera motion as object motion, resulting in suboptimal overall optimization that does not significantly surpass those of traditional 4D methods**. Furthermore, **Gao’s method is unable to achieve individual control over objects in multi-object reconstruction scenarios**, highlighting another key limitation compared to our approach.
>
> > Q6. Does the datasets used in the paper contain videos of dynamic objects and scenes? How does the baseline work?
> >
>
> Exactly, all our datasets[2][3] involve capturing dynamic objects with complex camera motions. Please refer to anonymous link in **Sec 9 of the supplementary** for more demos.
>
> In our experiments, we primarily compared three types of methods: static methods, 4D deformable methods, and controllable scene reconstruction methods. As for the static method like NeRF, the model is bound to fail and cannot complete the task, and we included them in the comparison to evaluate their reconstruction metrics for static backgrounds. Regarding the 4D methods, the model can control the scene in a temporal order along the time axis, but they cannot achieve control over individual objects. In contrast, the controllable methods require the model to not only model the static background accurately but also to control different objects independently. For a static scene, dynamic gaussian flows always become zeros, as illustrated in **Fig.3(a) of the manuscript**. In this case,  **Eq.(3)** degenerates to $\mathbf{u} = \mathbf{u}^\text{Cam}$ , which is a simpler setting that not only preserves algorithmic accuracy but also enhances stability. This observation further validates our insight into highlighting dynamic regions with dynamic Gaussian flow.
>
> We are deeply grateful for the reviewer's comments and hope that our response can clarify any misunderstandings regarding the motivation, novelty, and dataset, and look forward to a fair and objective evaluation.
>
> **Reference**
>
> [1] GaussianFlow: Splatting Gaussian Dynamics for 4D Content Creation
>
> [2] CoNeRF: Controllable Neural Radiance Fields
>
> [3] LiveScene: Language Embedding Interactive Radiance Fields for Physical Scene Rendering and Control
>
> [4] Subspace methods for recovering rigid motion I: Algorithm and implementation

---

> ### Author Response · Authors · 2024-11-29
> **Follow up process**
>
> Thanks for your time and effort in reviewing our work. We want to kindly follow up on the review process and greatly appreciate your feedback. Look forward to your response.

---

> ### Comment · Reviewer_jysV · 2024-12-03
> **Thanks for the response.**
>
> I've read the response provided by the authors.
>
> If the goal is motion decomposition and control, we could do it in 3D, e.g.,world space 3D motion =  camera space 3D motion + camera motion. This is part of existing dynamic 3D gaussian methods, where the 3D motion is typically represented in the world space (or the camera space of the first frame), given camera pose with regard to the scene is often known [A, B]. If I understand correctly, the introduced 2D motion decomposition, world space 2D motion = optical flow (which is camera space 2D motion) + projected 3D motion due to camera, is unnecessary and complicates the problem.
>
> While the response cleared my concerns about the evaluation and writing, this issue is unresolved and therefore I'm leaning towards a reject rating.
>
> [A] MoSca: Dynamic Gaussian Fusion from Casual Videos via 4D Motion Scaffolds.
>
> [B] Shape of Motion: 4D Reconstruction from a Single Video.

---

> ### Author Response · Authors · 2024-12-03
>
> Thank you for your thoughtful review. **The goal is to manipulate individual objects independently within the dynamic scene, not just to perform motion decomposition and control.**
>
> **Independent object control requires identifying the association between static background and dynamic objects for manipulation**. Hence, we employ optical flow prior to establishing this association, which captures dynamically changing objects in the scene. By motion decomposition, optical flow is decoupled into object motion and camera ego motion, isolating the influence of static background Gaussians. The object motion derived from optical flow represents all moving objects in the scene and is further separated by clustering, ultimately achieving independent control through spherical vector controlling scheme in **Sec 3.3**.
>
> The mentioned 4D reconstruction methods[1][2] model Gaussian ellipsoids in 3D space and replay the motions of all Gaussians over time. However, **they fail to separate individual objects from the dynamic scene.** Directly using 3D Gaussian motion for control fundamentally cannot achieve independent object control.
>
> Besides, the expression “world space 2D motion = optical flow (which is camera space 2D motion) + projected 3D motion” is ambiguous. Instead, optical flow on the image plane is composed of the projected 3D motion and camera ego-motion, i.e., **optical flow = projected 3D motion flow + camera ego-motion flow**, which was first introduced in [3].
>
> Therefore, **2D optical flow decomposition is essential to achieve independent control**. We kindly suggest that the reviewer refer to **Fig.9** and **Fig.10 of the supplementary**, as well as the project demo in **Sec 9,** for visual demonstrations of the task and the proposed method's effectiveness.
>
> [1] MoSca: Dynamic Gaussian Fusion from Casual Videos via 4D Motion Scaffolds.
>
> [2] Shape of Motion: 4D Reconstruction from a Single Video.
>
> [3] Subspace methods for recovering rigid motion I: Algorithm and implementation

---

### Author Response · Authors · 2024-11-20

We sincerely thank reviewer **#jysV, #j2YT, #rRqp** and **#93CX** for their thoughtful and constructive comments and suggestions. We are delighted to see that the reviewers have acknowledged the contributions and effectiveness of our work. Reviewer **#jysV** commended the soundness of our idea of spatial clustering for control. Both reviewer **#j2YT** and **#rRqp** appreciated the innovative approach of decoupling optical flow and analytically deriving the dynamic Gaussian flow to perceive controllable objects. Reviewer **#93CX** emphasized the significance of reconstructing dynamic scenes from monocular videos and enabling motion editing.

We appreciate their efforts to improve our manuscript and acknowledge the value of their comments. However, we would like to respectfully clarify some points that may have been misunderstood, Here, we reiterate the key contributions of our paper, which we believe address the reviewers' concerns:

- Annotations Free Gaussian Splatting method for controllable scene reconstruction, which automatically explores interactable scene structures with flow priors, and restores scene interactivity without any pixel/object-level manual annotations.
- Analytically derive the dynamic Gaussian flow constraints via differential analysis with alpha composition, establishing a mathematical link among optical flow, camera flow, and dynamic Gaussian flow that resolves the complex scenes of simultaneously accounting for camera motion and object motion, with CUDA implementation.
- A 3D spherical vector controlling scheme, avoiding traditional complex 1D control variable calculations bypassing 3DGS trajectory as state representation, further simplifying and accelerating interactive Gaussian modeling.

Clarify misunderstandings:

- FreeGaussian focuses on reconstructing the controllable scene with complex camera movement and object motions, not 4D reconstruction which replays a dynamic scene along the time axis.
- Previous methods rely on masks or dense control variable labels of controlled regions and consequently fail to generalize in their absence.
- FreeGaussian represents a groundbreaking achievement in unsupervised, large-scale scenario reconstruction and control. That’s why our method is significant and innovative.
- Extensive experiments, on both synthesis and real datasets, demonstrate our state-of-the-art performance with no annotations.

Concerns about DBSCAN:

- DBSCAN is more robust to noise with outliner handling and more flexible without predefined cluster numbers, compared with widely used clustering methods, such as KMeans, Besides, other methods like MeanShift may converge to local optima depending on the cluster landscape and initial window locations. Fig.19 and Fig.20 of the supplementary shows the effectiveness of DBSCAN in our dataset.

In ICLR2025, we value the feedback from our reviewers and have carefully addressed their concerns in our revised submission in response to their comments. In the supplementary materials of our revised PDF, we provide additional experiments, which can be summarized as:

- More View Synthesis Comparison results on CoNeRF-Controllable dataset.
- Qualitative results on CoNeRF faces and DyNeRF datasets.
- Visualization of dynamic Gaussian flow, clustering results, and novel view synthesis under training conditions with incomplete moving region.
- Qualitative results on #seq001 of OmniSim with noisy camera poses.
- Failure cases due to fast motion and lighting variation.
- Comparison among KMeans, MeanShift and DBSCAN with varying parameters.

---

### Meta-Review · Area_Chair_dhv8 · 2024-12-24

**Metareview:**

This work focuses on reconstructing scenes from monocular videos and enabling controllable editing of Gaussian motion.  The contributions include a derivation of 2D scene flow (referred to as dynamic gaussian flow) rendering and an interactive control pipeline that relies on spatial clustering. The task is interesting and the results appear to be good. However, the authors have not addressed the main concerns of reviewer  #jysV. Given the high standard of ICLR, this paper cannot be accepted in its current form.

**Additional Comments On Reviewer Discussion:**

In the discussion period, the authors have provided detailed responses to the reviewers' comments. However, the responses have not addressed the concerns of  reviewer #jysV, who rates this paper as "Reject".

---

### Decision · Program_Chairs · 2025-01-22

Reject